



# Elements of future snowpack modeling – Part 2: A modular and extendable Eulerian–Lagrangian numerical scheme for coupled transport, phase changes and settling processes

Anna Simson[1,4], Henning Löwe[2], and Julia Kowalski[3,4]

[1]AICES Graduate School, RWTH Aachen University, Schinkelstr. 2a, 52062 Aachen, Germany
[2]WSL Institute for Snow and Avalanche Research SLF, Flüelastr. 11, 7260 Davos, Switzerland
[3]Computational Geoscience, University of Göttingen, Goldschmidtstr. 1, 37077 Göttingen, Germany
[4]Methods for Model-based Development in Computational Engineering, RWTH Aachen University,
Eilfschornsteinstraße 18, 52062 Aachen, Germany

**Correspondence:** Anna Simson (simson@aices.rwth-aachen.de)

**Abstract.** A coupled treatment of transport processes, phase changes and mechanical settling is the core of any detailed snowpack model. A key concept underlying the majority of these models is the notion of layers as deforming material elements that carry the information on their physical state. Thereby an explicit numerical solution of the ice mass continuity equation can be circumvented, although with the downside of virtual no flexibility in implementing different coupling schemes for densification, phase changes and transport. As a remedy we consistently recast the numerical core of a snowpack model into an extendable Eulerian–Lagrangian framework for solving the coupled non-linear processes. In the proposed scheme, we explicitly solve the most general form of the ice mass balance using the method of characteristics, a Lagrangian method. The underlying coordinate transformation is employed to state a finite-difference formulation for the superimposed (vapor and heat) transport equations which are treated in their Eulerian form on a moving, spatially non-uniform grid that includes the snow surface as a free upper boundary. This formulation allows us to unify the different existing viewpoints of densification in snow or firn models in a flexible way and yields a stable coupling of the advection-dominated mechanical settling with the remaining equations. The flexibility of the scheme is demonstrated within several numerical experiments using a modular solver strategy. We focus on emerging heterogeneities in (two-layer) snowpacks, the coupling of (solid–vapor) phase changes with settling at layer interfaces and the impact of switching to a non-linear mechanical constitutive law. Lastly, we discuss the potential of the scheme for extensions like a dynamical equation for the surface mass balance or the coupling to liquid water flow.

## 1 Introduction

The snow density is probably the most important prognostic variable of any snowpack model as, e.g., reflected by a focus on snow water equivalent in past snow model intercomparison projects (Krinner et al., 2018, and references therein). That said, it actually comes as a surprise when not even the detailed snowpack models, e.g., Crocus and SNOWPACK, explicitly state an ice mass conservation equation in their technical documentation (Brun et al., 1989, 1992; Lehning et al., 2002; Bartelt and Lehning, 2002). Only a more detailed inspection reveals how mass conservation is accounted for, namely rather indirectly by stating a settling law for individual layers and resorting to a "Lagrangian coordinate system that moves with the ice matrix" (Bartelt and Lehning, 2002) to translate the ice-phase deformation into a thickness evolution of the layers (Brun et al., 1989; Vionnet et al., 2012). While this procedure has been well established for a long time, it is without numerical ambiguities only in the absence of phase changes. In addition, this non-explicit nature

of the most important conservation law in snow makes it virtually impossible to isolate and advance the numerical core of a snowpack model as an encapsulated numerical scheme comprising all involved coupled non-linear partial differential equations.

This non-explicit treatment of snow density or ice mass continuity in snowpack models, e.g., SNOWPACK and Crocus, has to be contrasted to other existing work on densification, comprising both stand-alone numerical snow studies (Meyer et al., 2020) and the vast body of work on firn densification (Lundin et al., 2017). All of the latter models are built around an explicit formulation of the ice mass continuity equation. This conceptual difference renders a general comparison of firn and snow densification mechanisms (Lundin et al., 2017) difficult. For model intercomparisons in the future it is thus desirable to have a numerical core that is able to digest arbitrary snow or firn densification physics with a flexible but rigorous coupling to superimposed non-linear transport and phase change processes.

Any holistic snowpack model has to account for transport of heat, vapor and liquid water and its induced phase change processes, as well as mechanical settling and apparent metamorphic processes acting on the snow's microstructure. A widespread body of literature exists that proposes different modeling approaches and computational tools for the various flavors and perspectives of this multi-physics coupled situation, e.g., Krinner et al. (2018, and references therein). For the general timescales of interest (diurnal up to seasonal), it is common practice to employ a continuum assumption and to model the snowpack's state as a mixture of ice, vapor, water and air, as initially described in Bader and Weilenmann (1992). Detailed snowpack models, such as SNOWPACK (Lehning et al., 2002) and Crocus (Vionnet et al., 2012), are built upon this type of mixture theory approach and used for a wide range of purposes.

While heat transport, mechanical settling and processes due to the presence of liquid water have been incorporated into SNOWPACK (Bartelt and Lehning, 2002) and Crocus (Vionnet et al., 2012) for a long time, effects due to vapor transport have mostly recently been investigated in separate studies. Temperature gradients between the ground and atmosphere imply upward vapor fluxes in snowpacks. Stronger temperature gradients (due to either a smaller snowpack height or colder surface temperature) in arctic conditions yield higher vapor fluxes (Domine et al., 2019) compared with alpine snowpacks. Depth hoar layers with reduced density and thermal conductivity form at the snowpack's bottom. In alpine snowpacks similar hoar layers develop within the snowpack that may cause avalanches due to their low mechanical stability (Schweizer et al., 2003). Upscaled and homogenized continuum mechanical process models that account for vapor transport have been put forward, for instance by Hansen and Foslien (2015) and Calonne et al. (2014). Both couple the snowpack's evolving temperature profiles to a non-linear reaction–diffusion type of equation for va-

por transport and phase change. While they provide different flavors of how to set up the underlying mathematical model, both approaches are formulated for idealized conditions and investigate vapor diffusion in the absence of settling and therefore neglect its feedback on the apparent snow density. These model-based investigations and also field-based observations in arctic snowpacks on top of permafrost (Domine et al., 2016, 2019) have demonstrated the significance of vapor-related processes in snow. Hence, it is of great interest to investigate further how vapor interacts with apparent mechanical processes within the snowpack.

Incorporating vapor transport directly into a fully coupled snowpack model is however challenging, e.g., due to the fact that the associated characteristic timescales are small, and expected effects on the snowpack are localized (Schürholt et al., 2021). To resolve these processes on small timescales and at specific locations requires a much higher spatio-temporal resolution than is typically provided by existing operational schemes. In its original version, SNOWPACK for instance uses time steps on the order of 15 min or longer (Bartelt and Lehning, 2002) to facilitate seasonal simulation times. For Crocus time steps are on the order of 15 min (Viallon-Galinier et al., 2020) to 1 h (Vionnet et al., 2012). The recent work of Jafari et al. (2020) provides a first attempt to account for vapor transport within a coupled snowpack model. In their paper, they accounted for diffusive vapor transport and phase change following Hansen and Foslien (2015) and analyzed its feedback on the snow density. In order to resolve diffusive processes, simulations were conducted at much shorter time steps of 1 min and a finer spatial resolution of approximately 0.1 cm. For comparison, a typical layer thickness in SNOWPACK is 2 cm (Wever et al., 2016) and the minimum layer thickness in Crocus is 0.5 cm (Brun et al., 1989). While the work of Jafari et al. (2020) demonstrates the general feasibility of vapor-coupled snowpack models, the exact nature of how vapor transport and phase changes interfere with stress-induced settling remains to be investigated in depth.

It is well known that any numerical strategy that aims at simulating simultaneous settling-induced deformation of the snowpack and (arbitrary) diffusive transport requires a special computational treatment to couple both. Diffusive transport and reactive phase change are best modeled by taking an Eulerian perspective, hence on a static mesh. In contrast there exist a number of different techniques to incorporate the settling-induced deformation. One option is to use a time-dependent coordinate transformation by Morland (1982), who developed a fixed domain transformation to solve one-phase diffusion problems with a moving free surface on a finite, time-invariant computational domain. An alternative approach was put forward by Wingham (2000), who used a different spatio-temporal coordinate transformation for firn densification. Both transformation strategies effectively eliminate the vertical motion (or gradients of it) from the computational update procedure. And exactly the same is

(implicitly) performed in the present treatment of densification in snowpack models (Bartelt and Lehning, 2002; Vionnet et al., 2012), where the coordinate transformation embodied in the deformation of the underlying computational grid through the update of layer positions and/or thicknesses is (implicitly) exploited for the ice mass conservation. However, the present descriptions do not take full advantage of a clear and explicit separation into a Lagrangian deformation module that accounts for mechanical settling and an Eulerian transport and phase change module. The benefit of this hybrid computational strategy is that it is easy to understand, computationally feasible, provides a modular error control and increases the interpretability by disentangling numerical artifacts from features of the underlying non-linear process models. Hybrid numerical schemes that combine an Eulerian process model with a Lagrangian-type spatio-temporal mesh adaptation are not new. These schemes have been used in other disciplines, e.g., for phase change problems (Lacroix and Garon, 1992); as $\sigma$ coordinates in oceanography, where the ocean's surface and bottom are projected onto coordinates $\sigma = 0$ and $\sigma = -1$ that follow the ocean floor's topography (Mellor and Blumberg, 1985); or for shallow-flow models (Kowalski and Torrilhon, 2019).

The aim of our work is twofold: first, we will describe our numerical strategy for a phase-changing snowpack. The numerical scheme is hybrid, in the sense that it clearly discriminates between a solution of the mechanical settling operator by means of a Lagrangian approach and a solution to the transport and phase change operator by means of an Eulerian approach. To some degree, the numerical model description must be understood as a rigorous re-formulation of the numerical schemes from existing computational snowpack models. Yet, in addition to existing schemes we (a) explicitly separate the Eulerian and Lagrangian part of the solver to facilitate a later modular adaption, (b) provide a full finite-difference formulation including correction terms due to the deforming (non-uniform) mesh that are typically omitted, and (c) discuss options to increase the approximation accuracy of the various parts of the numerical scheme. Second, we demonstrate the computational potential by applying and analyzing simulation results for an idealized two-layer, dry-snow situation. We consider a model cascade of different process building blocks, which in their most comprehensive version, correspond to fully coupled heat and vapor transport alongside phase changes and settling.

With this work we seek to contribute to anticipated future developments of snow or firn models or likewise extensions of existing ones that aim at flexibility and modularity while providing a simple, mathematically rigorous numerical approximation for a stable and robust integration of generic multi-physics process equations. By modularity and extendability we understand the possibility of considering or neglecting specific process modules and parametrizations in a straightforward way. This modularity would enable us to (a) investigate competing non-linear effects systematically

**Table 1.** Terminology of state variables, model parameters and constants.

| Symbol | Name | Equation/value | Unit |
|---|---|---|---|
| State variables | | | |
| $\phi_i$ | Ice volume fraction | Eq. (1) | – |
| $\rho_v$ | Vapor density | Eq. (A1) | $kg\,m^{-3}$ |
| $T$ | Temperature | Eq. (11) | K |
| Model parameters of snow | | | |
| $v$ | Vertical velocity | Eq. (7) | $m\,s^{-1}$ |
| $c$ | Ice deposition rate | Eq. (9) | $kg\,m^{-3}\,s^{-1}$ |
| $\dot{\epsilon}$ | Strain rate | Eq. (3) | $s^{-1}$ |
| $\eta$ | Viscosity | Eq. (4) | $Pa\,s$ |
| $\sigma$ | Stress | Eq. (5) | $Pa\,m^{-2}$ |
| $\rho_{snow}$ | Density | Eq. (5) | $kg\,m^{-3}$ |
| $D_{eff}$ | Vapor diffusion coefficient | Eq. (A2) | $m^2\,s^{-1}$ |
| $(\rho C)_{eff}$ | Heat capacity | Eq. (A4) | $J\,m^{-3}\,K^{-1}$ |
| $k_{eff}$ | Thermal conductivity | Eq. (A3) | $W\,m^{-1}\,K^{-1}$ |
| Parameters assumed to be constant (Calonne et al., 2014) | | | |
| $\rho_i$ | Ice density $\phi_i = 1$ | 917 | $kg\,m^{-3}$ |
| $L$ | Ice latent heat of sublimation | 2 835 333 | $J\,kg^{-1}$ |
| $C_i$ | Ice heat capacity $\phi_i = 1$ | 2000 | $J\,kg^{-1}\,K^{-1}$ |
| $\rho_a$ | Air density $\phi_i = 0$ | 1.335 | $kg\,m^{-3}$ |
| $C_a$ | Air heat capacity $\phi_i = 0$ | 1005 | $J\,kg^{-1}\,K^{-1}$ |

from a cascade of process models, (b) assess the quality of the numerical approximation independently and (c) conduct a standardized model selection based on well-defined benchmarks.

The paper is structured as follows. In Sect. 2, we recall the dry-snow model equations comprising the relevant transport, phase change and mechanical aspects. In Sect. 3, we introduce the Eulerian–Lagrangian numerical scheme and its solution using the method of characteristics. In Sect. 4, we present and discuss results from a number of simulation scenarios, including verification scenarios that consider transport, phase changes and mechanics in the absence of any interaction, as well as coupled scenarios that focus on their interplay. We furthermore investigate the impact of different viscosity parametrizations and assess the behavior when switching to a Glen type of non-linear constitutive closure. Finally, we compare our results to a conventional layer-based treatment. In Sect. 5, we summarize and discuss our findings, and in Sect. 6 we draw conclusions regarding future snowpack modeling.

## 2 Physical model

### 2.1 General situation

As a common starting point, snow models take a macroscale perspective that volume averages (Bartelt and Lehning, 2002; Bader and Weilenmann, 1992; Hansen and Foslien, 2015) or homogenizes (Calonne et al., 2014) the snowpack's microstructural state into macroscale variables. If not stated

otherwise, we implicitly assume all state variables to be macroscale variables. State variables, model parameters and constants used in this paper are summarized in Table 1.

In the most general case, snow is a mixture of ice, air, vapor and water, and the snow density is given as a mixture of the respective pure densities (Bader and Weilenmann, 1992; Morland et al., 1990). The amount of ice in one reference volume of snow is $\phi_i \rho_i V$, in which $\phi_i$ denotes the ice's volume fraction, $\rho_i$ its pure density and $V$ the volume of the reference volume. For dry snow, further contributions due to water vapor can be neglected, and the snow density can be approximated as $\phi_i \rho_i$. The structure and volume fraction of the ice can change over time either due to strain-induced settling processes or due to transient phase changes, such as sublimation and deposition or melting and freezing. Our paper focuses on the derivation of a hybrid Eulerian–Lagrangian framework to solve settling, transport and phase changes with an assessment of the computational building blocks. To this end we restrict ourselves to dry snow and allow for one secondary phase (vapor) in an Eulerian treatment coupled to the Lagrangian treatment of the ice phase. With respect to computational model development, we regard the dry-snow situation as the more challenging (yet less investigated) one compared to the wet-snow situation, mostly due to a broader spectrum of characteristic spatial and temporal scales involved (see more detailed discussion in Sect. 6).

Note, that water transport and solid–liquid phase change can in principle be integrated following a similar strategy to that presented in this paper. The following section introduces the (macroscale) snowpack model where the subsection structure reflects the later-described modular structure of the numerical core.

## 2.2 Ice mass balance

The ice volume fraction $\phi_i = \phi_i(z, t)$ within a spatio-temporally evolving snowpack of varying snow height $H(t)$ is governed by the ice mass balance and reads

$$\partial_t \phi_i + \nabla \cdot (v \phi_i) = \frac{c}{\rho_i}, \tag{1}$$

with velocity field $v$, source term $c$ and ice density $\rho_i$ (Hansen and Foslien, 2015; Bader and Weilenmann, 1992). Note that mechanical settling is neglected in Part 1 of this companion paper. The corresponding ice mass balance (Eq. 7 in Part 1) does thus not include the velocity field $v$.

In a 1D situation that focuses on an evolving vertical snow column, we have the vertical position $z$ as the only relevant spatial coordinate ($z \in [0, H(t)]$). The velocity field $v$ reduces to vertical velocity $v = v(z, t)$, which depends on time and the position within the column. It is negative for snow height decrease and positive for snow height increase. Vertical motion results either from mechanical settling, hence a consolidation or compaction of the snowpack, or alternatively as a continuity response to changes in ice volume from

sublimation, deposition, melting and freezing via the source term $c$. The continuity response leads to a minor vertical decrease/increase in snow height. Though effects due to consolidation of snow may be significantly more pronounced than those due to phase change processes in the pore space, the latter needs to be accounted for to acknowledge mass conservation of the complete system. At this point in time, we do not consider any additional increases in snow height due to precipitation, yet we discuss how this can be included in the future in Sect. 5.

The source term $c = c(z, t)$ varies with time and position in the column and stands for a gain or loss of ice mass from phase change (Bader and Weilenmann, 1992) per unit volume and unit time. As we constrain this paper to the dry situation we will henceforth refer to $c$ as the deposition rate. $c$ is positive (production) if new ice is built, namely vapor deposits, and it is negative (loss) if ice is lost, namely sublimates. Finally, $\rho_i$ denotes the constant pure density of ice and serves as a scaling factor. The ice mass balance (Eq. 1) couples mechanical settling and phase change processes. Considering the equation in its full form is essential for our goal to model and eventually analyze the interplay between these processes. The structure of the ice mass balance resembles an advection–reaction equation that can conveniently be solved by means of Lagrangian-type computational methods, such as the method of characteristics (see Sect. 3). Yet in order to do so, we need to provide a closure for both vertical velocity $v$ and deposition rate $c$.

## 2.3 A closure for the velocity field

Velocity $v$ represents mechanical deformation in the snowpack. Its idealized relation to the strain rate is given by

$$\nabla v = \dot{\epsilon}. \tag{2}$$

Note that this is simplified with respect to more general, tensorial formulations of 1D consolidation theories; see for instance Audet and Fowler (1992). Yet even the idealized formulation Eq. (2) will be sufficient for our purposes, as it resembles the approach implicitly chosen in snowpack models (Bartelt and Lehning, 2002; Vionnet et al., 2012).

In general one would expect that porous snow inherits the non-linear constitutive behavior of ice (Kirchner et al., 2001), which leads to

$$\dot{\epsilon} = \frac{1}{\eta} \sigma^m, \tag{3}$$

which is a variant of Glen's law. Here, $\eta$ denotes the compactive viscosity of snow and $\sigma$ denotes the stress. The choice of the Glen exponent $m$ in earlier work depends on both the physical regime and the computational feasibility. The linear form of Glen's law ($m = 1$) is chosen in Vionnet et al. (2012) and Bartelt and Lehning (2002). For the sake of comparability we thus mainly use a linear version of Glen's law; hence $m = 1$. Our framework, however, also copes with

the non-linear relation, such as $m = 3$, and we later include a comparative example.

The compactive viscosity $\eta$ depends on the snow's microstructure and is challenging to determine from experiments (Wiese and Schneebeli, 2017). It is typically provided as a parametrized closure for a specific physical situation and strongly correlates with the choice for the Glen exponent $m$. This fact clearly constrains its universal applicability and makes any transfer of a validated snowpack model to other physical situations challenging. In this article, we will consider both constant-viscosity scenarios, as well as an additional scenario with a varying viscosity assuming an empirical viscosity closure from Vionnet et al. (2012)

$$\eta(\phi_i, T) = f \eta_0 \frac{\rho_i \phi_i}{c_\eta} \exp(a_\eta(T_{ph} - T) + b_\eta \rho_i \phi_i), \quad (4)$$

with the state variables temperature $T$ and ice volume fraction $\phi_i$; the constants ice density $\rho_i$ and phase change temperature $T_{ph} = 273\,\text{K}$; and further constants $\eta_0 = 7.62237 \times 10^6\,\text{kg}\,\text{s}^{-1}$, $a_\eta = 0.1\,\text{K}^{-1}$, $b_\eta = 0.023\,\text{m}^3\,\text{kg}^{-1}$ and $c_\eta = 250\,\text{kg}\,\text{m}^{-2}$. Finally, $f$ reflects properties of the snow microstructure, i.e., the angularity and the size of the grains, and it is assumed to be 1 in our case. The constant-viscosity value applied to a linear Glen's law $\eta_{\text{const},m=1}$ is derived with intermediate values for the ice volume fraction and temperature of the respective initial conditions. These values are plugged into the empirical closure Eq. (4) to solve for viscosity. The same procedure cannot be applied to derive a constant-viscosity value for the non-linear version of Glen's law $\eta_{\text{const},m=3}$ since the viscosity closure (Eq. 4) was initially calibrated to the linear form of Glen's law ($m = 1$). Instead we choose a snow deformation rate from the literature ($\dot{\epsilon} = 10^{-6}\,\text{s}^{-1}$; Johnson, 2011) and determine the maximum stress value from the initial snow density. These strain rate and stress values are then inserted into the constitutive relation (Eq. 3), which is finally solved for viscosity. To avoid infinite ice volume growth above physical values ($\phi_i > 1$), the viscosity must tend to infinity for $\phi_i \to 1$. Therefore, the constant-viscosity values are restricted to ice volumes below 0.95 by multiplication with an ice-volume-fraction-dependent power law (Appendix Eq. A6). This power law yields $\sim 1$ for $\phi_i \leq 0.95$ and exponentially increases for higher ice volumes. Multiplied with the constant-viscosity values, viscosity remains constant below $\phi_i < 0.95$ and exponentially increases above it, which stops further densification and settling. This procedure does not intend to reproduce the correct physics for low-porosity ice, although it mathematically leads to a similar crossover behavior.

In the absence of strong horizontal deformation and deviatoric stress components, it is reasonable to assume a stress-free condition at the snow's surface and a hydrostatic stress condition in its interior:

$$\nabla \sigma = g \rho_{\text{snow}} . \quad (5)$$

$g$ is the gravitational acceleration, and $\rho_{\text{snow}}$ refers to the snow's density, which is clearly dominated by the ice fraction via $\rho_{\text{snow}} \approx \phi_i(z)\rho_i$. It varies with the position $z$ in the snow column due to a vertically varying ice volume fraction $\phi_i(z)$. Integration of Eq. (5) and combination with Eqs. (2) and (3) yields an expression for the velocity gradient:

$$\partial_z v = \frac{1}{\eta} \left( g \int_z^{H(t)} \phi_i(\zeta)\rho_i \mathrm{d}\zeta \right)^m . \quad (6)$$

$\zeta$ is the integration variable. A second integration along the vertical axis finally yields an expression for the velocity at position $z$ in the snow column:

$$v(z) = - \int_0^z \frac{1}{\eta} \left( g \int_{\tilde{z}}^{H(t)} \phi_i(\zeta)\rho_i d\zeta \right)^m \mathrm{d}\tilde{z} , \quad (7)$$

in terms of total height $H(t)$ and the ice volume fraction $\phi_i(z,t)$ and with $v(z = 0, t) \equiv 0$. This definition of the vertical velocity yields a process that complies with the obvious physical constraints: (a) the velocity vanishes at the bottom of the snow column, hence preventing artificial penetration into the ground. This is similar to displacement requirements in SNOWPACK (Bartelt and Lehning, 2002). (b) The vertical velocity accumulates with height, which prevents any artificial disaggregation of the snowpack. (c) The vertical velocity relaxes towards zero as the ice volume fraction tends towards its maximum volume fraction $\phi_i < \phi_{i,\text{max}} < 1$. In the remainder of this paper, we will use Eq. (7) to account for the mechanical settling of the snowpack.

## 2.4 Transport and phase changes

The ice deposition rate $c$ as relevant to solve Eq. (1) typically depends on a cascade of coupled heat and mass transport for the involved phases of ice, water and vapor. In this article, we will consider a process model proposed by Hansen and Foslien (2015) that reflects a dry-snow condition in which void space is filled by vapor only. Note, however, that this coupled process model could readily be substituted or extended by another one, e.g., from Calonne et al. (2014), Jafari et al. (2020) or Schürholt et al. (2021).

Next, we state the essential aspects and process equations of the model proposed in Hansen and Foslien (2015) and describe how it can be used to recover the ice deposition rate.

Assuming a dry-snow condition, the ice production is solely determined by mass transport between vapor and ice. The vapor mass balance reads

$$\partial_t(\rho_v(1 - \phi_i)) - \nabla \cdot (D_{\text{eff}} \nabla \rho_v) = -c, \quad (8)$$

in which $\rho_v$ denotes the vapor density and $D_{\text{eff}}$ the effective vapor diffusion coefficient. Vapor production corresponds to a negative ice deposition rate $-c$ that represents sublimation. Following Hansen and Foslien (2015), vapor density in

the pore space can be assumed to be at saturation density $\rho_v^{eq}$, so $\rho_v \equiv \rho_v^{eq}$. The latter is well investigated, and empirical relations exist that specify its temperature dependency $\rho_v^{eq}(T)$. In this work, we will employ an empirical relation from Libbrecht (1999). The full expression can be read in Appendix A1. Due to the closure for vapor density $\rho_v^{eq}$, the vapor mass balance (Eq. 8) can be rewritten using the temperature dependence of the equilibrium vapor density:

$$(1 - \phi_i)\frac{d\rho_v^{eq}}{dT}\partial_t T - \nabla \cdot \left( D_{eff}\frac{d\rho_v^{eq}}{dT}\nabla T \right) = -c. \tag{9}$$

Assuming the snow to be in thermal equilibrium at the microscale, we can likewise write the energy balance in terms of the temperature, which reads

$$(\rho C)_{eff}\partial_t T - \nabla \cdot (k_{eff}\nabla T) = cL. \tag{10}$$

The parameters $(\rho C)_{eff}$ and $k_{eff}$ stand for the effective heat capacity of snow and effective thermal conductivity, respectively. Both parameters depend on the ice volume fraction, and their definition is stated in Appendix A2. The right-hand side of the heat equation (Eq. 10) accounts for latent heat release, which is coupled to phase change processes.

The system of the two equations, Eqs. (9) and (10), and the two unknowns, temperature $T$ and deposition rate $c$, is solved by replacing $c$ in Eq. (10) with Eq. (9), which yields a non-linear equation for temperature:

$$\left( (\rho C)x_{eff} + (1 - \phi_i)\frac{d\rho_v^{eq}(T)}{dT}L \right)\partial_t T$$
$$= \nabla \cdot \left( \left( LD_{eff}\frac{d\rho_v^{eq}(T)}{dT} + k_{eff} \right)\nabla T \right). \tag{11}$$

The spatio-temporal temperature evolution is then used to recover the ice deposition rate $c$ from either Eq. (9) or Eq. (10).

## 3 Computational approach

The complete process model is now given by the ice mass balance Eq. (1), its mechanically induced vertical velocity Eq. (7), and the coupled system for temperature Eq. (11) and ice deposition rate determined by either Eq. (9) or Eq. (10). Each of the equations will be solved in a separate module. The ice mass balance in conjunction with the vertical velocity has the form of a non-linear advection equation, whereas the remaining equations are of parabolic nature, which is reflected in our general approach to solve the system.

### 3.1 General approach to the computational strategy

Based on the distinction into diffusion- and advection-dominated processes, we propose a two-step solution scheme:

*Step 1*. This step accounts for the mesh deformation and solves the advection-dominated mechanical settling, i.e., the ice mass balance Eq. (1), by means of a Lagrangian approach that tracks the movement of the coordinates including changes from metamorphism.

*Step 2*. This step determines the spatio-temporal evolution of temperature and deposition rate fields as introduced in Sect. 2.4 based on an Eulerian approach that solves the diffusion-dominated transport and phase changes via a finite-difference implementation on a deformed (unstructured) mesh.

Note that here we employed a finite-difference method because it provides a feasible algorithm that is applicable to the scenarios considered in the paper using a 1D snow column. It also naturally integrates with the Lagrangian part of the solution (Step 1), as we can re-use the same mesh. In principle, it is also possible to couple the two-step approach with a finite-element solution for the temperature and deposition rate, for instance when aiming for a 2D or 3D model in a complex geometry that incorporates realistic mountain slope topographies. When using a finite-element solver, however, we have to keep in mind that the deposition rate and temperature fields need to be reconstructed from the solution at each time step. Especially when wanting to use higher-order elements, this might limit computational feasibility.

Our solution scheme alternates both steps via straightforward first-order operator splitting. This is found to work well for our simulation scenarios yet could be readily exchanged with a higher-order splitting scheme, e.g., a second-order Strang splitting (LeVeque, 2002), if required.

The computational model is implemented in Python, and it is modular and extendable, in the sense that each module can be separately activated and deactivated. This not only simplifies the verification of individual process building blocks but also allows an in-depth investigation of the various coupling effects and the model's non-linear feedback. Alternative formulations e.g., of the parametrized velocity field are implemented and can easily be exchanged. Finally, the modular structure facilitates the implementation of additional closure relations or the integration of entire new process modules.

### 3.2 Computational grid

In this paper, we consider a 1D snow column, which is discretized into $nz+1$ spatial mesh nodes denoted by $z_k$ with $k \in \{0, 1, \ldots, nz\}$. We applied 101 computational nodes ($nz = 100$) except for some simulations that required a higher resolution of 251 nodes ($nz = 250$). The mesh is non-uniform in general, meaning that the distance between neighboring nodes $z_{k+1} - z_k$ varies throughout the snow column and with time. Note that the $z$ axis is oriented opposing gravitational acceleration, such that $z_0$ denotes the position of the ground and $z_{nz}$ the position of the snowpack's free surface. Time increments are denoted by $t_n$ with $n \in \{0, 1, \ldots, nt\}$ and $nt$ being the maximum number of time steps in a complete sim-

ulation run. For each of the field variables subscript $k$ denotes the vertical coordinate and superscript $n$ denotes the time step; hence $T(z_k, t_n) = T_k^n$.

## 3.3 Lagrangian solution of the ice mass balance

When the snowpack is subject to vertical motion, e.g., settling, its physical height decreases; hence its vertical extent shrinks. One option to reflect this in a computational method is to adjust the spatial node coordinates accordingly. The challenging fact in our situation is that the vertical motion within the snow column (non-linear advection) is coupled to phase changes, i.e., a change in the ice volume fraction via the source term in the ice mass balance (Eq. 1). The method of characteristics is a suitable method to solve such a non-linear advection equation with source terms. It can be interpreted as a simultaneous motion tracking of snow reference volumes, referred to as the integration along so-called characteristics, while also accounting for its metamorphism along the trajectory. By construction, the method correctly tracks the snowpack's moving free surface. Due to the fact that the snow column's evolution is determined with respect to a reference volume that moves vertically at speed $v$ in the snowpack, the method of characteristics is called a Lagrangian approach.

In order to derive the specific update rule for the ice mass balance Eq. (1), we first apply the product rule to its initial Eulerian version

$$\partial_t \phi_i + v \partial_z \phi_i = \frac{1}{\rho_i} c - \phi_i \partial_z v \tag{12}$$

and then re-formulate the equation in a Lagrangian reference frame, hence with respect to nodes moving at the vertical velocity $v$. Changing to the moving reference frame effectively compensates for the advection term in Eq. (12) and yields

$$\partial_t \phi_i = \frac{1}{\rho_i} c - \phi_i \partial_z v, \tag{13}$$

$$\partial_t z = v. \tag{14}$$

Equation (14) accounts for the settling of material particles within the snowpack. We will use it to update the coordinates of the mesh nodes directly, which results in a continuous mesh deformation as illustrated in Fig. 1. Equation (13) captures the evolution of the ice volume fraction along the trajectory of a moving ice volume within the snowpack. It accounts for volume changes due to (a) mass production and loss in response to phase changes and (b) vertical variation in the vertical velocity. Further details and generalizations of the method of characteristics can be found in Farlow (1993).

Equations (13) and (14) can be solved analytically for a constant vertical velocity and deposition rate. In our case however, the velocity closure is provided by Eq. (7) and the deposition rate results from solving yet another process

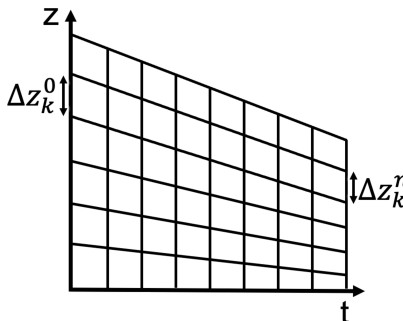

**Figure 1.** Computational mesh. The snowpack height varies with time, e.g., shrinks due to settling of the snow. This has to be incorporated into the computational mesh, which undergoes deformation due to the downward movement of the free surface. The initially equidistant mesh does not uniformly change, which results in a mesh of varying node distances, so in general $\Delta z_k^0 \neq \Delta z_k^n$ and $\Delta z_k^n \neq \Delta z_{k+1}^n$.

model (Eqs. 10 and 11), which requires a numerical solution. Since we expect the response of the ice volume fraction to be slow (with respect to other processes in the system), we will rely on a first-order explicit Euler time integration scheme:

$$\phi_{i,k}^{n+1} = \phi_{i,k}^n + \Delta t^n \left( \frac{1}{\rho_i} c_k^n - \phi_{i,k}^n \partial_z v_k^n \right), \tag{15}$$

$$z_k^{n+1} = z_k^n + \Delta t^n v_k^n . \tag{16}$$

In order to update the mesh coordinates according to Eq. (16) for the vertical velocity closure derived before, we need to numerically approximate Eq. (7) at each node $z_k$, which results in

$$v(z_k) = -\sum_{j=0}^k \left( \frac{1}{\eta} \sigma_j^m \right) \Delta z_j^n , \tag{17}$$

with $\Delta z_j^n := z_{j+1}^n - z_j^n$ where $j \in [0, nz)$, $m$ being the Glen exponent, $\eta$ being viscosity and $\sigma_j$ denoting the stress exerted by the overburdened snow mass

$$\sigma_j = \sum_{l=j}^{nz} g \phi_{i,l} \rho_i \Delta z_l^n , \tag{18}$$

where $g$ is gravitational acceleration. Note that the stress at the uppermost node $k = nz$ is zero, so velocity $v(z_{nz})$ is only controlled by the movement below and is thus equivalent to the velocity at the next lower node ($v(z_{nz-1})$). The forward Euler scheme of Eqs. (15) and (16) via the method of characteristics combined with the velocity update (Eq. 17) essentially resembles the treatment of mass conservation as it is, for instance, presently carried out in SNOWPACK. However, the explicit formulation and numerical treatment of Eqs. (15) and (16) allows us to also employ other (e.g., higher-order, implicit) solution schemes for both equations if this is required to capture detailed aspects of the spatio-temporal coupling of phase changes ($c$) and settling (via $\partial_z v$) (cf. also

the discussion in Sect. 5). To solve Eq. (15), we directly discretize the velocity's spatial derivative $\partial_z v$, which corresponds to the strain rate $\dot{\epsilon}_k^n = \frac{1}{\eta}\sigma_k^m$ given via Eq. (3). This is beneficial, as it avoids numerically approximating the velocity gradient. The complete numerical update of ice volume fraction $\phi_i$ and mesh coordinates $z$ can now concisely be written as

$$\phi_{i,k}^{n+1} = \phi_{i,k}^n + \Delta t^n \left( \frac{1}{\rho_i} c_k^n + \frac{1}{\eta}\left( \sum_{l=k}^{nz} g\phi_{i,l}^n \rho_i c, \Delta z_l^n \right)^m \phi_{i,k}^n \right),$$
(19)

$$z_k^{n+1} = z_k^n + \Delta t^n \left( \sum_{j=0}^{k} \frac{1}{\eta}\left( \sum_{l=j}^{nz} g\phi_{i,l}^n \rho_i \Delta z_l^n \right)^m \Delta z_j^n \right).$$
(20)

Similarly to existing layer-based schemes (see for instance Sect. 3.4. in Bartelt and Lehning, 2002, or its recent extension in Jafari et al., 2020), the method of characteristics provides information on the settling of layers within the snowpack. Yet, in addition, it serves as a basis for a fully modular and flexible computational strategy that (a) by construction accounts for the two-way feedback between the ice volume fraction and mass production or decay rates resulting from phase changes as a response to transport processes within the snowpack and (b) allows for a flexible adoption and extension of the process model (used to determine $c$) and the velocity closure. The latter could for instance serve as a pathway to integrate a data-driven velocity closure (or assimilation) from measurements. Such flexibility in numerical tools will be important in the future to conduct model comparisons, such as presented in Schürholt et al. (2021) within holistic snowpack models, or even a formalized Bayesian model selection that allows for inferring the most plausible process model out of a pool of candidate models given certain data. A remaining difficulty now is to provide a (Eulerian) numerical scheme for diffusive processes that can operate on a spatially varying unstructured mesh.

### 3.4 Eulerian solution of transport and phase changes on a moving mesh

The process model accounting for vapor transport and heat transport (Eqs. 9 and 11) has to be solved with respect to a moving computational mesh according to Eq. (16). Both equations have the same generic structure; namely

$$\alpha \partial_t T - \partial_z (\beta \partial_z T) = \gamma,$$
(21)

with $\alpha = \alpha_T = (\rho C)_{\text{eff}} + (1 - \phi_i)\frac{d\rho_v^{\text{eq}}(T)}{dT}L$, $\beta = \beta_T = k_{\text{eff}} + LD_{\text{eff}}\frac{d\rho_v^{\text{eq}}(T)}{dT}$ and $\gamma = \gamma_T = 0$ for the heat equation (Eq. 11) and $\alpha = \alpha_c = (1 - \phi_i)\frac{d\rho_v^{\text{eq}}(T)}{dT}$, $\beta = \beta_c = D_{\text{eff}}\frac{d\rho_v^{\text{eq}}(T)}{dT}$ and $\gamma = \gamma_c = -c$ for the vapor transport equation (Eq. 9).

An implicit first-order finite-difference approximation of Eqs. (9) and (11) for a spatially varying mesh of increments

$\Delta z_k^n$ results in

$$\alpha_{T,k}^n \frac{T_k^{n+1} - T_k^n}{\Delta t^n}$$
$$= \frac{2\beta_{T,k}^n \left( \left( T_{k+1}^{n+1} - T_k^{n+1} \right) - \left( T_k^{n+1} - T_{k-1}^{n+1} \right) \right)}{\left( \Delta z_k^n \right)^2 + \left( \Delta z_{k-1}^n \right)^2}$$
$$+ \frac{\beta_{T,k+1}^n - \beta_{T,k-1}^n}{\Delta z_k^n + \Delta z_{k-1}^n} \frac{T_{k+1}^{n+1} - T_{k-1}^{n+1}}{\Delta z_k^n + \Delta z_{k-1}^n}$$
$$+ E_T \left( T_{k+1}^{n+1}, T_{k-1}^{n+1} \right),$$
(22)

$$\alpha_{c,k}^n \frac{T_k^{n+1} - T_k^n}{\Delta t^n}$$
$$= \frac{2\beta_{c,k}^n \left( \left( T_{k+1}^{n+1} - T_k^{n+1} \right) - \left( T_k^{n+1} - T_{k-1}^{n+1} \right) \right)}{\left( \Delta z_k^n \right)^2 + \left( \Delta z_{k-1}^n \right)^2}$$
$$- c_k^{n+1} + \frac{\beta_{c,k+1}^n - \beta_{c,k-1}^n}{\Delta z_k^n + \Delta z_{k-1}^n} \frac{T_{k+1}^{n+1} - T_{k-1}^{n+1}}{\Delta z_k^n + \Delta z_{k-1}^n}$$
$$+ E_c \left( T_{k+1}^{n+1}, T_{k-1}^{n+1} \right).$$
(23)

Note that parameters $\alpha_f$ and $\beta_f$ for $f \in \{T, c\}$ also vary in space and time and will be (explicitly) evaluated based on the snowpack's state at time $n$. The terms $E_c$ and $E_T$ are higher-order mesh errors for the vapor and temperature equations. These higher-order mesh errors account for the necessary correction due to non-uniformity of the mesh and are controlled by the temperature gradient; they vanish for equidistant meshes or constant temperatures. The complete form of the higher-order mesh errors is given in Appendix B, and their effect on the accuracy of the simulation is discussed in Sect. 4.3.

The complete numerical update can be concisely written in matrix form, which matches with the way it is implemented in the software:

$$T^{n+1} = (\mathbf{A}_{T+E_T})^{-1}\left( \mathbf{B}_T T^n \right),$$
(24)
$$c^{n+1} = (\mathbf{A}_c + \mathbf{E}_c)T^{n+1} + \mathbf{B}_c T^n.$$
(25)

First, Eq. (24) is solved for temperature $T^{n+1}$. Next, the updated temperature is used to solve Eq. (25) for the deposition rate $c^{n+1}$. The complete matrix definitions are given in Appendix C. Note that, formally, it would be possible to add up matrices $\mathbf{A}_T$ and $\mathbf{E}_T$ as well as $\mathbf{A}_c$ and $\mathbf{E}_c$. We decided to keep them in this particular form to stress the similarity of this formulation with a standard finite-difference approximation on an equidistant mesh, in which we are left with $\mathbf{B}_T$ and $\mathbf{B}_c$ and $\mathbf{E}_T$ and $\mathbf{E}_c$ vanish.

### 3.5 Iterative coupling of Eulerian and Lagrangian solutions

The derived numerical update routines for temperature, deposition rate, vertical velocity and ice volume fraction com-

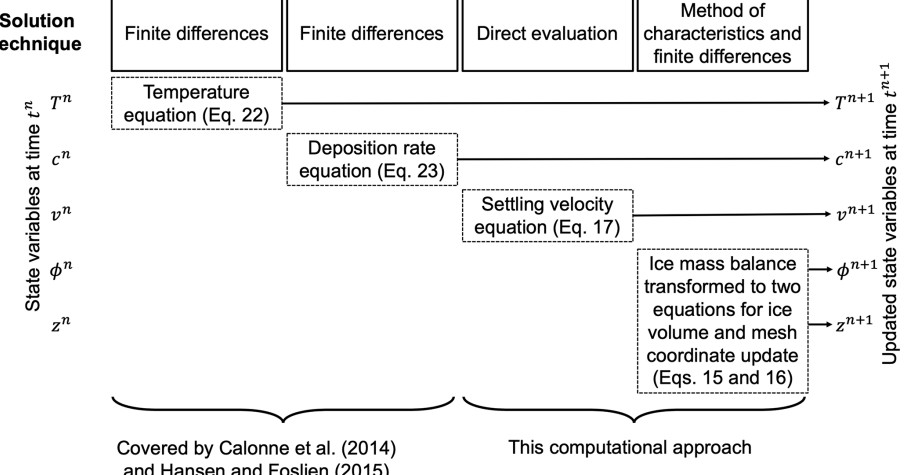

**Figure 2.** The computational workflow of one iteration. The state variables at time $t^n$, depicted on the left-hand side, are updated through the modules annotated as dashed-outline boxes that are ordered diagonally in the center of the figure. After each update the state variables at time $t^{n+1}$ are retrieved. The equations of the modules are implemented into the computational model through the respective solution technique stated in the solid-outline boxes in the top row. The computational steps are carried out from top to bottom. The iterative approach can be summarized as (1) determine time step size $\Delta t$ according to Eq. (26), (2) update the temperature field based on Eq. (22), (3) compute the deposition rate with the temperature field based on Eq. (23), (4) determine the vertical velocity with Eqs. (17) and (18), and (5) update the ice volume fraction and the mesh coordinates simultaneously based on Eqs. (19) and (20). While (2) and (3) are a re-implementation of an existing approach previously published by Hansen and Foslien (2015) and Calonne et al. (2014), their coupling to (4) and (5) constitutes the novelties of our work. Note that (4) is computed as part of (5) in the code.

prise the four main modules that are sequentially called to update the respective state variables for one time step. A schematic illustration is given in Fig. 2. The equations for heat and vapor transport have already been implemented by Calonne et al. (2014) and Hansen and Foslien (2015). A feedback on the ice volume fraction in the absence of a vertical velocity has been investigated in Part 1 of the companion paper (Schürholt et al., 2021). The modules for vertical velocity and the coupled update of ice volume fraction and mesh coordinates, through the method of characteristics, are novel in our approach. Our implementation is modular in the sense that it allows for a coupling with other process models that comply with a non-uniform mesh.

The time step size for the next time step $n + 1$ is dynamically updated in the computational scheme. Since diffusive processes are dominant, we utilize the mesh Fourier number based on the diffusivity $\frac{\beta_T}{\alpha_T}$ of heat of the current time step $n$:

$$\Delta t^{n+1} = \min_k \left( \frac{0.5\alpha_{T,k}^n \left(\Delta z_k^n\right)^2}{\beta_{T,k}^n} \right). \tag{26}$$

Since this choice for the time step computation did not yield instabilities, we excluded the vapor's diffusivity for the time step computation. Note that in response to settling processes, the mesh sizes vary and decrease (see Fig. 1) with time, and so does the time step.

In the following, we describe how the modularity of the model is applied and used to assess the individual effect of

the different process building blocks by a strategical activation and deactivation of the modules.

### 3.6 Application of the model

We applied the developed numerical scheme to perform several simulations with varying combinations of *activated* and *deactivated* advection- and diffusion-type process building blocks, e.g., transport and phase changes, such as those also considered in Schürholt et al. (2021), vs. transport and phase changes in the presence of settling and their corresponding coupled scenarios. Furthermore, this scheme allowed the numerical verification of separate building blocks. While the scenarios are still idealized, they demonstrate the robustness of the Eulerian–Lagrangian scheme against the selection of varying sub-sets of model components. Table 2 provides an overview of the various combinations we considered as they have been introduced in Sect. 3. Note that we use the terms vertical velocity and settling velocity interchangeably.

Firstly, we focus on the effects due to pure mechanical settling on the snowpack (Case 1). Next, we consider isolated heat transport (Case 2) as well as its interplay with settling processes (Case 3). Similarly, we consider coupled heat and vapor transport first in the absence of settling (Case 4) and later with settling (Case 5). For Case 5, we evaluate the effect of included or excluded higher-order mesh errors $E_T$ and $E_c$ (see Sect. 3.4) on the temperature profiles. Case 1, Case 3 and Case 5 consider the constant viscosity for a linear Glen's law ($m = 1$) $\eta_{const,m=1}$, as introduced in Sect. 2.3. Further-

**Table 2.** List of the various simulation scenarios, referred to as *cases*, in which we activate different combinations of process building blocks and consider constant-viscosity and non-constant-viscosity closures. Heat transport induces vapor transport and triggers phase changes. Cases 5, 7 and 8 are also referred to as *fully coupled processes*.

| Case | Heat transport (Eq. 22) | Vapor transport (Eq. 23) | Mechanics (Eqs. 19 and 20) | | | |
|---|---|---|---|---|---|---|
| | | | Viscosity (Sect. 2.3) | | Glen's law (Eq. 17) | |
| | | | $\eta = \text{const}$ | $\eta(\phi_i, T)$ | $m = 1$ | $m = 3$ |
| Case 1 | | | ✓ | | ✓ | |
| Case 2 | ✓ | | | | | |
| Case 3 | ✓ | | ✓ | | ✓ | |
| Case 4 | ✓ | ✓ | | | | |
| Case 5 | ✓ | ✓ | ✓ | | ✓ | |
| Case 6 | | | | ✓ | ✓ | |
| Case 7 | ✓ | ✓ | | ✓ | ✓ | |
| Case 8 | ✓ | ✓ | ✓ | | | ✓ |

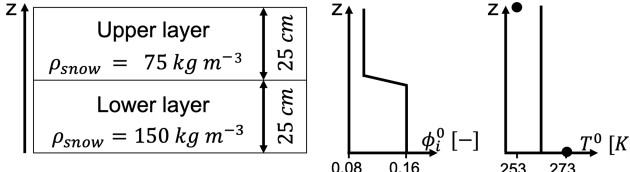

**Figure 3.** The initial condition of the snowpack regarding snow density on the left-hand side and profile plots of the initial ice volume fraction ($\phi_{i,0}$) and temperature ($T_0$) on the right-hand side. There are two snow layers with equal thickness of 25 cm, yielding a snowpack of 50 cm height. The bottom layer has a higher density of $150\,\text{kg}\,\text{m}^{-3}$, and the upper layer's density is $75\,\text{kg}\,\text{m}^{-3}$. The $z$ axis of the 1D model increases in an upward direction, so $z = 0$ denotes the ground. Downward-directed movements are thus described by negative velocities. The vicinity of the interface between the two layers is referred to as the transition area. The initial ice volume fraction is derived from the initial snow density. Its profile ($\phi_{i,0}$) shows the linear decrease over 2 cm of the ice volume fraction in the transition area from the lower to the upper layer. The initial temperature profile ($T_0$) is constant at 263 K. The black dots mark the constant temperature boundary conditions: 273 K at the bottom and 253 K at the top.

more, we investigate the impact of an empirical, temperature-controlled and ice-volume-fraction-controlled viscosity closure (Eq. 4), first on settling only (Case 6) and then on the fully coupled processes (Case 7). Next, we show that our general approach can be combined with the non-linear Glen's law (Eq. 3) by using $m = 3$ (Case 8) and an accordingly adjusted constant viscosity $\eta_{\text{const},m=3}$. For a detailed explanation of the general derivation of the viscosity values see Sect. 2.3. Lastly, we compare our new modeling approach to that of layer-based schemes (cf. Sect. 1).

## 3.7 Computational setup, initial and boundary conditions

*Initial condition.* The initial ice volume fraction $\phi_i$ reflects a layered situation as depicted in Fig. 3, with two snow layers of equal thickness. The bottom layer has an initial snow density of $150\,\text{kg}\,\text{m}^{-3}$, and the upper layer's density is $75\,\text{kg}\,\text{m}^{-3}$. The transition from the upper layer to the lower layer is linearly smoothed out over 2 cm, which for a grid defined according to Sect. 3.2 corresponds to 5 computational nodes for the coarser and 11 computational nodes for the finer discretization. The snow densities are in the range of "damped new snow" and "new snow", respectively (Paterson, 1994). Snow densities in this range are expected, e.g., for new snow in the European Alps (Helfricht et al., 2018) or in the Rocky Mountains (Judson and Doesken, 2000). We choose this layered snowpack to ensure an extreme and very active snow regime with a strong dynamical coupling of the processes. Temperature is initially constant at 263 K throughout the whole snowpack. The deposition rate is directly deduced from temperature (see Eq. 23) and therefore requires neither initial nor boundary conditions. From the initial condition we derived the constant-viscosity values $\eta_{\text{const},m=1} \approx 9.1 \times 10^7\,\text{Pa}\,\text{s}$ and $\eta_{\text{const},m=3} \approx 16 \times 10^{12}\,\text{Pa}\,\text{s}$; see also Appendix A3.

*Boundary condition.* We consider a constant temperature of 273 K at the bottom boundary and a constant temperature of 253 K at the free surface.

*Simulation time.* We simulate 2 d (48 h), 3 d (62 h) and 4 d (96 h) scenarios.

## 4 Results and discussion

### 4.1 Settling (Case 1)

First, we investigate the effects of mechanical settling on the snowpack (Case 1 in Table 2) and in particular the evolution of the vertical velocity (Fig. 4a) and the ice volume fraction

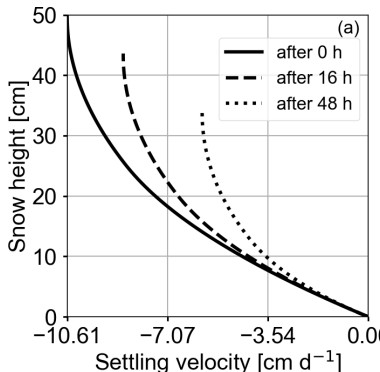
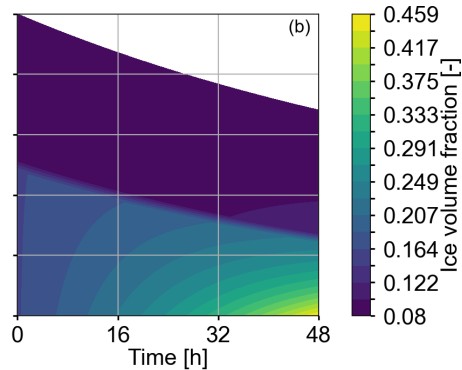

**Figure 4.** Plots show the results of Case 1 of Table 2, corresponding to isolated settling effects. Panel **(a)** depicts vertical velocity over snow height. The profiles show the state of the snowpack at initiation, after 16 and after 48 h. It is clearly visible that vertical velocity and absolute snow height decrease with time. Panel **(b)** depicts the ice volume fraction over time. We interpret the lower, lighter part of the plot, where changes in the ice volume fraction are clearly discernible, as the lower layer and the darker, upper part of the plot as the upper layer. Changes in ice volume fraction are visible in the lower layer; it increases the most at the bottom of the snowpack.

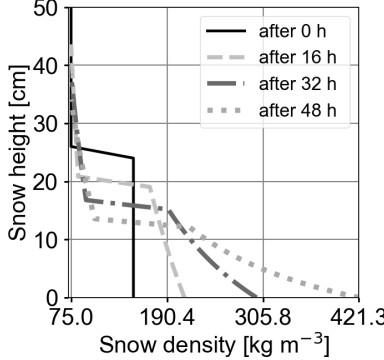

**Figure 5.** Snow density profiles ($x$ axis) over snow height ($y$ axis) at initiation and after 16, 32 and 48 h. Snowpack height decreases and snow density increases with time at all locations, but they do so the most in the lower part of the snowpack.

(Fig. 4b). The vertical velocity decreases from top to bottom and relaxes during the first 48 h. Vertical velocity varies more in the lower layer compared to in the upper layer within one time step. This pattern remains prominent as time proceeds while the overall velocity variation decreases. This effect is due to the increase in the overburdened snow mass from top to bottom. Settling proceeds the fastest just after the start of the simulation, when the snowpack is at maximum height, and correspondingly its snow density is the lowest. In the course of time the ice volume fraction increases faster in the lower layer than in the upper layer, and it is the highest at the bottom of the snowpack (Fig. 4b). This observation is also visualized in Fig. 5, which depicts profiles of snow density. Furthermore, the extent of the upper layer decreases only slightly, approximately 3.5 cm, over the simulation time, whereas the lower layer reduces to half of its initial height (approximately 12.5 cm). These effects are expected and reflect the correlation between the amount of compaction and the total overburdened mass. The total settlement of 14 cm after 2 d means a 30 % snow height reduction. Bartelt and Christen (2007) simulated an 11.6 to 54.8 cm snow height reduction after 5 d for an initially 90 cm high snowpack of 115 kg m³ density, which is a snow height reduction of 12 % to 60 %. Taking into account that snow settles more slowly with increasing density, our results fit to the highest settling rate derived by Bartelt and Christen (2007).

## 4.2 Heat transport in the absence and presence of settling (Cases 2 and 3)

In this subsection, we first consider isolated heat transport. This simulation scenario refers to Case 2 in Table 2. Temperature (Fig. 6a) and temperature gradients (Fig. 6c) reach a stationary state after approximately 60 h. Heat flux differences between the two layers are clearly visible in the temperature gradient plot. Next, heat transport is superposed by mechanical settling (Fig. 6b and d), representing Case 3. As a result, snow height decreases while the internal temperature profiles evolve. Active mechanical processes yield a steepened temperature gradient and hence a higher value of the heat flux (Fig. 6d). This effect can be attributed to

- the decrease in snow height while keeping the temperatures at the boundaries fixed and

- the permanent change in thermal conductivity and thermal diffusivity due to their dependency on variations in the ice volume (Eq. A3).

The temperature profile will reach the stationary state once the ice volume fraction has reached its maximum value.

https://doi.org/10.5194/tc-15-1-2021

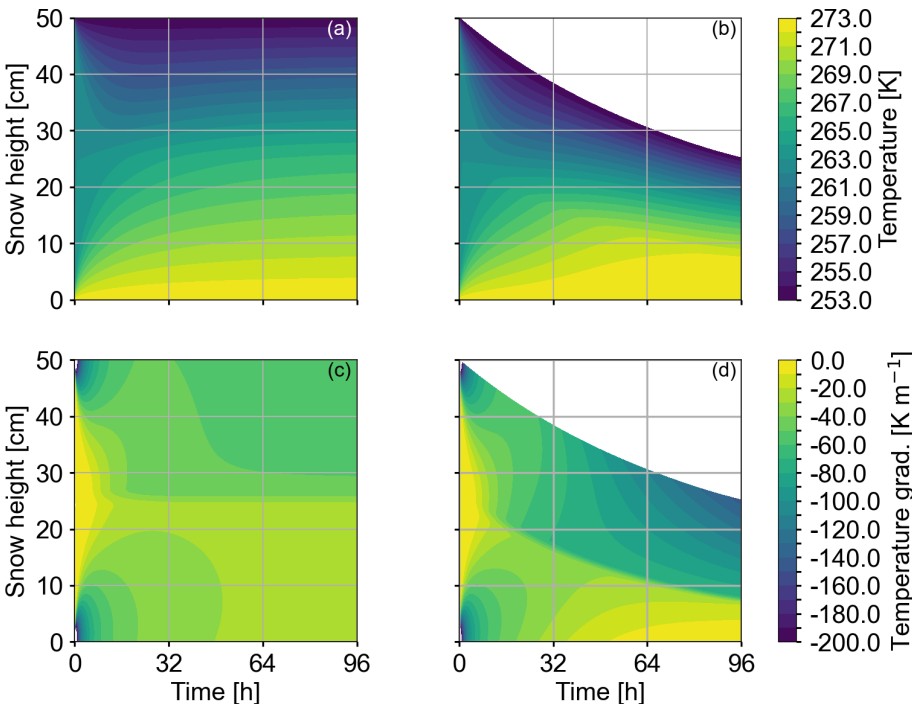

**Figure 6.** Panels **(a)** and **(c)** show the results for Case 2 of Table 2 corresponding to heat transport being solely active. Panels **(b)** and **(d)** show the results for Case 3 of Table 2 corresponding to active heat transport and mechanical settling. For each plot the $y$ axis represents snow height and $x$ axis time. The plots in the top row (**a**, **b**) show the temperature evolution, and the plots in the bottom row (**c**, **d**) show the respective temperature gradients. In **(a)** and **(c)** settling is inactive, so the boundary of upper layer and lower layer is at the snowpack center. In **(b)** and **(d)**, we interpret the upper, darker part, which is characterized by higher gradients, as the upper layer and the lower, lighter area with lower gradients as the lower layer. The initial conditions for both cases are equivalent (see Fig. 3). In Case 2 the temperature profile **(a)** has reaches the stationary, piecewise linear profile after approximately 60 h. In Case 3 the temperature profile **(b)** is not yet stationary at the end of the simulation (96 h) as mechanical processes are still yielding a change in the ice volume fraction. The temperature gradient **(c)** will become constant only when the maximum ice volume fraction has been reached.

## 4.3 Heat and vapor transport in the absence and presence of settling (Cases 4 and 5)

By using the vapor formulation from Hansen and Foslien (2015), transport of vapor through and phase changes in the snowpack both require an apparent temperature gradient such that the evolution of vapor transport can only be considered in conjunction with heat transport. In Fig. 7a, we compare the deposition rate (negative for sublimation) due to heat and vapor transport only (Case 4 in Table 2) with the deposition rate obtained when considering additional settling processes, representing the fully coupled processes (Case 5 in Table 2). Both profiles are characterized by moderate deposition rates throughout the snow column with a pronounced negative (sublimation) peak at the center of the snow column, which is located in the transition area of the layers. The profile for the fully coupled processes shows a higher sublimation peak (approximately 4 times higher). Figure 7b shows the time evolution of the fully coupled processes (Case 5). In the first hours, sublimation is low in the transition area. After approximately 6 h, the pronounced sublimation rate peak, as already described for (Fig. 7a), develops and increases until

the end of the simulation (48 h). The increased sublimation in the layer transition area may be driven by strong vapor density gradients (Fig. 7c) above the transition area that can be inferred from a strong, local temperature gradient (Fig. 7d). This temperature gradient is further enhanced (Fig. 6d) by compaction due to settling for Case 5, which yields even stronger variations in the material properties in the transition area than without compaction and explains the stronger sublimation rates for the fully coupled processes.

Furthermore, both profiles in Fig. 7a show a small peak in the deposition rate (positive $x$ direction) just above the aforementioned sublimation rate peak. This peak is very weak for Case 4 and more prominent for Case 5. This deposition rate peak is highly interesting as it is interpreted as the onset of spatio-temporal oscillations as observed and investigated in greater detail in the companion paper (Schürholt et al., 2021). Schürholt et al. (2021) describe these wiggles as "smooth oscillations" that are "intrinsic features" of the equations. The results in Fig. 7a nicely demonstrate that (a) our Eulerian–Lagrangian scheme can capture this behavior and (b) the instability prevails and even increases in the presence of settling processes. The results suggest that mechanics likely in-

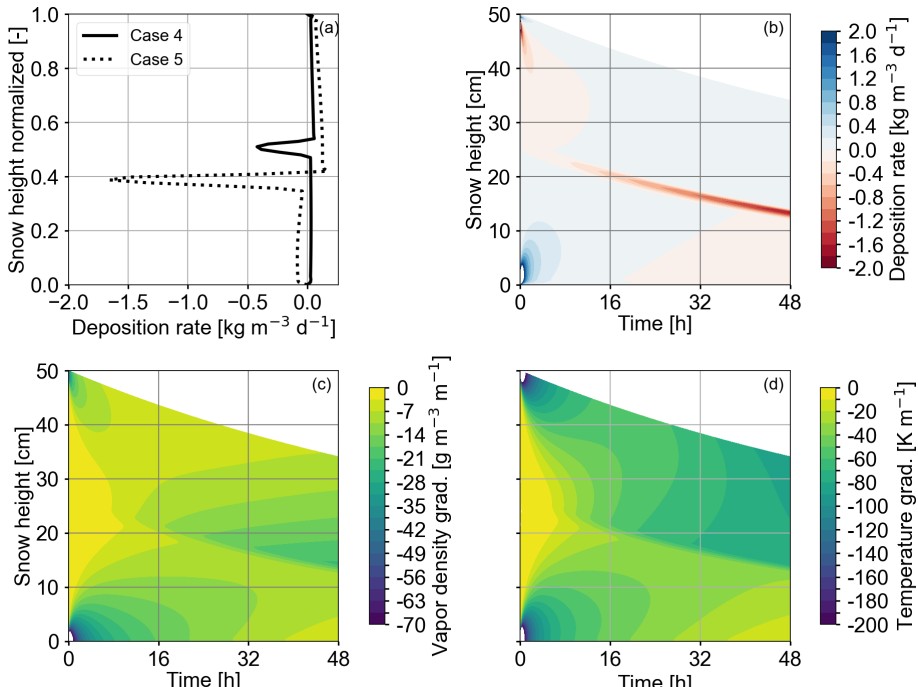

**Figure 7.** Panel **(a)** shows two deposition rate profiles over the normalized snow height after 2 d. The solid line represents the results of heat and vapor transport in the absence of settling for Case 4 of Table 2. The dashed line refers to Case 5 of Table 2, which additionally accounts for settling. Sublimation rates (negative deposition rates) for Case 5 (fully coupled processes) are increased by approximately a factor of 4 with respect to Case 4 without settling. At the top of the sublimation peak for both cases, a slight peak in the deposition rate is visible. Panel **(b)** shows the deposition rate profile evolution for Case 5. A pronounced sublimation rate peak in the transition area is first visible after approximately 6 h and increases with time. We interpret this area of increased sublimation (red line in the center) as the boundary of the upper and lower layer. Panel **(c)** shows the evolution of the vapor density gradient. The gradient at the bottom of the upper layer (at approximately 20 and 15 cm height after 16 and 48 h) increases with time. Panel **(d)** shows the evolution of the temperature gradient with time. Overall the temperature gradient is higher in the upper compared to in the lower layer. The lobes at the top and bottom at the start of the simulation in **(b–d)** are due to the strong phase change activity and heat flux triggered by the initial and boundary conditions.

crease local phase change activity in the vicinity of layer boundaries, which potentially has a large effect on weak layer formation.

The deposition rates obtained with our model are between $-2$ and $2\,\mathrm{kg\,m^{-3}\,d^{-1}}$, which fits to the range of $-1.728$ to $1.728\,\mathrm{kg\,m^{-3}\,d^{-1}}$ presented in Jafari et al. (2020). Sublimation rate peaks on the order of 0.1 to $1.2\,\mathrm{kg\,m^{-3}\,d^{-1}}$ have also been computed with the numerical test cases by Hansen and Foslien (2015). For comparison with experiments, deposition rates can be derived via $\mathrm{SSA}\cdot v_{\mathrm{n}}\cdot\rho_{\mathrm{i}}$, with $v_{\mathrm{n}}$ being the ice crystal's interface growth velocities and SSA the ice's specific surface area (see Calonne et al., 2014, Eq. 21 therein). For a simple characteristic scale analysis, we considered SSA in the range of $0.6\times 10^{4}$ to $1\times 10^{4}\,\mathrm{m^{-1}}$ (Schleef et al., 2014) and an interface growth velocity on the order of $1\times 10^{-9}\,\mathrm{m\,s^{-1}}$ (Krol and Löwe, 2016; Calonne et al., 2014). Combining these literature values yields deposition rates on the order of $0.5\times 10^{4}\,\mathrm{kg\,m^{-3}\,d^{-1}}$, which is significantly larger than our simulation results. Interface growth velocities on the order of $10^{-13}$ or $10^{-14}\,\mathrm{m\,s^{-1}}$ would match with the simulated magnitudes for the deposition rate.

Lastly, we evaluate the impact of included higher-order mesh errors $E_{\mathrm{T}}$ and $E_{\mathrm{c}}$ (see Sect. 3.4) on the temperature distribution. We determine the error by computing the temperature deviation between the solution that considers higher-order mesh errors and the solution that does not. The deviation is then quantified in an L1 norm. The error increases with simulation time and is 0.13 K after 24 h, 0.23 K after 36 h and 0.28 K after 48 h. After 48 h the deviation is highest for the computational nodes just above the layer transition, where high temperature gradients are present (see Fig. 6). Note that the error for the deposition rate could be derived similarly. From the temperature error, the deposition rate error can be derived as the deposition rate is directly derived from temperature via the vapor transport equation; we consider one error measure as sufficient to emphasize the impact of mesh errors.

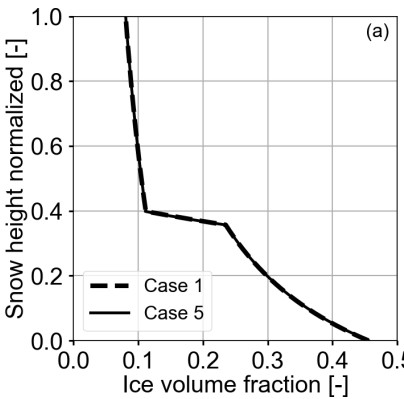 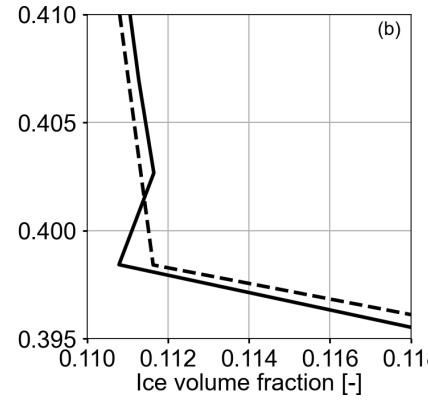

**Figure 8.** The plots show ice volume fraction profiles over the normalized snow height after 2 d. **(a)** Case 1 (Table 2) depicts the ice volume fraction corresponding to solely active settling, and Case 5 refers to the fully coupled processes. Panel **(b)** zooms in to the density transition area of **(a)**. The kink in the profile of Case 5 shows the effect of the increased sublimation in Fig. 7 that yields a local decrease in the ice volume fraction. In order to better resolve the kink of Case 5, we increased the number of grid nodes to 251.

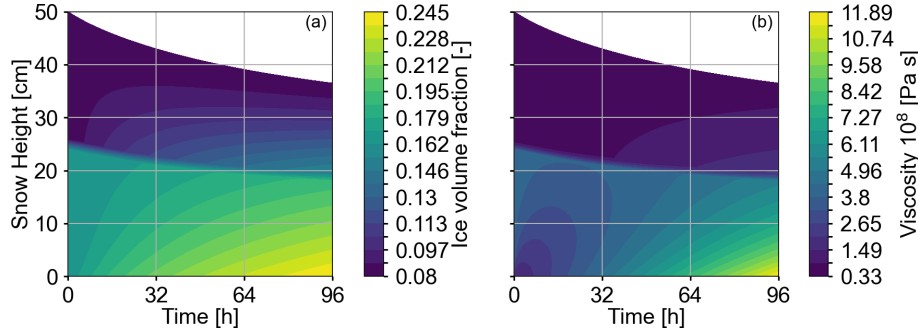

**Figure 9.** The plots show the evolution of the ice volume fraction **(a)** and viscosity **(b)** for 4 d for Case 7 of Table 2, which refers to the fully coupled processes combined with a dynamically varying viscosity. Snow height is depicted on the $y$ axis. The upper and lower layers are interpreted in both plots as the darker areas in the upper part and the lighter areas in the lower part. In **(a)** ice volume fraction increases most at the bottom of the upper layer. The lower layer consolidates less than the upper layer. In **(b)** viscosity increases more slowly in the upper layer and increases faster (by up to 1 order of magnitude) in the lower layer.

## 4.4 Settling-induced evolution of the ice volume fraction in the absence and presence of transport (Cases 1 and 5)

In this section, we compare isolated settling (Case 1 in Table 2) and the fully coupled processes (Case 5 in Table 2) with respect to their impact on the evolving ice volume fraction. Figure 8 shows the corresponding ice volume fraction profiles after 2 d. Both profiles are very similar (Fig. 8a), which suggests that the density evolution is dominated by settling processes and coupled heat and vapor transport play a minor role. When focusing on the upper boundary of the transition area (Fig. 8b), we find however a locally decreased ice volume fraction for the fully coupled processes (Case 5). This suggests a local ice volume decay for active vapor transport and implies phase changes. This observation is consistent with the enhanced sublimation rate observed in Fig. 7 and indicates the formation of a density heterogeneity.

## 4.5 Heat and vapor transport coupled to settling with a dynamic viscosity (Cases 4, 6, and 7)

Figure 9 shows the evolution of the ice volume fraction and viscosity over 4 d for Case 7 (Table 2), which is the fully coupled processes coupled to dynamic viscosity. The ice volume fraction increases gradually throughout the snow column (Fig. 9a). In Fig. 9b, we see that the viscosity of the upper layer has smaller values, and they also increase more slowly compared to viscosity values in the lower layer. In contrast, viscosity increases by approximately 1 order of magnitude in the lower layer. This derives from the applied viscosity formula that is controlled by the variables of temperature and ice volume fraction. Based on the formula, viscosity varies more with respect to ice volume fraction changes than to temperature changes. The ice volume fraction varies more in the lower layer, which then also yields more variation in viscosity. Additionally, the height of the lower layer decreases less than that of the upper layer. This outcome may be related to

The Cryosphere, 15, 1–24, 2021

https://doi.org/10.5194/tc-15-1-2021

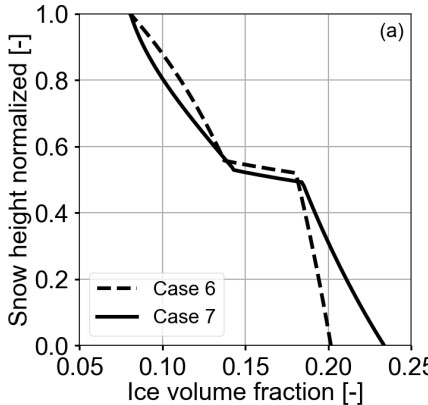 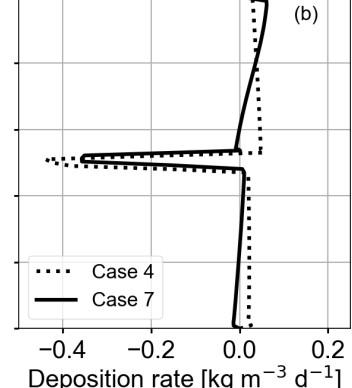

**Figure 10.** Panel **(a)** shows the ice volume fraction over the normalized snow height after a 3 d simulation time obtained with a dynamic viscosity. Case 6 (Table 2) corresponds to mechanical settling solely, and Case 7 represents the fully coupled processes. Panel **(b)** shows the deposition rate over the normalized snow height for Case 7 and Case 4 after a 3 d simulation time. Case 4 represents heat and vapor transport in the absence of settling. For the fully coupled processes (Case 7) the sublimation rate peak at the layer transition is slightly lower compared with inactive settling (Case 4). Both peaks are approximately at the same location of the normalized snow height. Case 7 shows reduced deposition rates in the area above the peak compared with Case 4. Case 7 also has a small peak in the deposition rate just above the sublimation rate peak.

the lower layer's higher viscosity (higher resistance to deformation).

Figure 10a compares the ice volume fraction profiles after a 3 d simulation time of Case 7 to those of Case 6 (fully coupled processes, non-dynamic viscosity). For the dynamic viscosity, the ice volume fraction is higher in the lower layer and lower in the upper layer compared to that of constant viscosity. This is due to the dynamic viscosity's temperature dependence. Temperatures at the bottom are close to the melting point and yield lower viscosities. Thus, settling proceeds faster and compaction is stronger in the lower part. The opposite is true for the upper layer. Since we used an intermediate value of 263 K to derive the constant viscosity, variations in the center of the snowpack are less pronounced. Figure 10b shows the deposition rate of Case 7 compared with Case 4, which refers to deactivated settling. Similarly to Fig. 7 both deposition rate profiles have a sublimation rate peak in the transition area. Dissimilar to Fig. 7b is that the peaks are approximately at the same normalized snow height and that the peak of the fully coupled processes is not higher than the one of deactivated settling. Instead Case 7 shows less deposition in the vicinity of the transition area above the sublimation rate peak compared to Case 4. This suggests that the sublimation rate peak is less pronounced when coupled to the proposed dynamic viscosity, but settling still has an effect on the deposition rate. Additionally, Case 7 shows the small peak in deposition rate just above the sublimation rate peak, which is similar to Case 4 and discussed in Sect. 4.3.

### 4.6 Non-linear Glen's law in a fully coupled dry-snowpack model of constant viscosity (Case 8)

In this simulation scenario, we present the results of the fully coupled processes for the non-linear Glen's law (Eq. 3 with $m = 3$, Case 8). As discussed before (Sect. 2.3), the viscosity closure (whether it is a constant value or an empirical closure) strongly depends on the choice of the Glen parameter $m$. This requires us to adjust the constant-viscosity value accordingly; see Sect. 2.3 for details.

Figure 11a (Case 8 in Table 2) shows vertical velocity profiles and the evolution of the ice volume fraction with time. The vertical velocity is almost constant in the upper layer and then decreases in the lower layer. This effect is similar to the vertical velocity profiles as presented and explained for the linear version of Glen's law (Fig. 4a), but it is more pronounced due to the non-linearity in the constitutive law. Compared with previous scenarios, the overall vertical velocity is lower. This is probably related to the magnitude of the constant viscosity and cannot be directly related to the non-linear constitutive law. A further sensitivity study in the future would be most informative.

In Fig. 11b the upper layer's ice volume fraction and thickness remain almost constant with time. In contrast, the lower layer decreases 9 cm in height while the ice volume fraction increases with time from top to bottom.

As shown for Case 7 (Fig. 7a) the deposition rate profile shows a sublimation peak in the layer transition area (Appendix D1) that increases with time. Overall, however, deposition rates tend to be lower compared to those of preceding computations. The reduced phase change activity in the layer transition area can be directly related to smaller vertical variations in the temperature profile. This effect may be due to

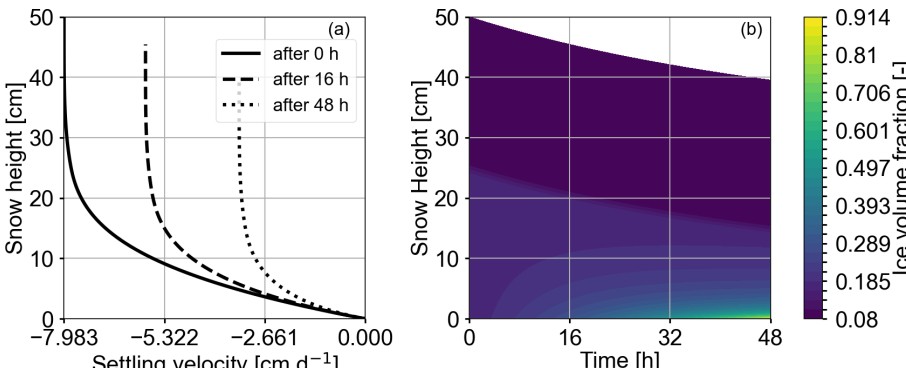

**Figure 11.** The plots show vertical velocity profiles over snow height **(a)** and the evolution of the ice volume fraction **(b)** for Case 8 of Table 2, which refers to the fully coupled processes combined with a non-linear Glen's law. Velocity **(a)** varies less compared with a linear version of Glen's law, which yields a more uniform evolution of the snowpack's ice volume fraction **(b)**. In **(b)** we interpret the locations of the upper layer as the darker area with an almost constant extent in the upper part and that of the lower layer as the slightly lighter area below.

less variation in the vertical velocities that yield a more uniform deformation and a less pronounced variation in the ice volume fraction across the layer transition area.

### 4.7 Comparison against layer-based schemes (based on Case 6)

In this section, we compare results of our proposed Eulerian–Lagrangian scheme with conventional layer-based models. We would like to emphasize that a two-layer snowpack model certainly constitutes an extremely simplified case, as layer-based schemes are usually operated with a significantly higher number of snow layers. Yet it is informative to conduct this analysis to point out differences, as these can certainly accumulate during long simulation times.

In layer-based snowpack models state variables are assigned layerwise, and the two-layer snowpack (Fig. 3) would have three computational nodes at the following locations: at the bottom of the lower layer, at the top of the lower layer and at the top of the upper layer. The two nodes located at the top of the lower and upper layers would then represent the physical state of the lower and the upper layer, respectively. Velocity is again derived from stress exerted by the overburden snow mass. Since the upper layer is represented by the computational node at the top, it is unloaded and requires a special treatment for stress. We adopt the approach by Vionnet et al. (2012) and apply a "non-physical stress" equivalent to half of the layer's own weight, yet we apply it to the uppermost computational node (Sect. 3.4 in Vionnet et al., 2012). Next, vertical velocity is computed likewise with Eq. (7) and viscosity with Eq. (4). We compare both approaches based on Case 6 of Table 2, hence in the presence of mechanical settling and for a dynamic viscosity closure. Since we neglect heat and vapor transport, the viscosity changes over time are solely due to the evolution of the ice volume fraction alone.

In Fig. 12, we see that the layer-based scheme sustains a layerwise vertical velocity (Fig. 12a) and ice volume fraction

evolution (Fig. 12c): one value for the velocity and one value for the ice volume fraction describe an entire layer. In contrast, using the generalized Lagrangian approach described in Sect. 3, we yield a sublayer resolution of the vertical velocities (Fig. 12b) and ice volume fractions (Fig. 12d). For both approaches (layer-based and Eulerian–Lagrangian) the vertical velocity is higher in the top part of the snowpack and zero at the bottom. For early times, the layer-based scheme determines a vertical velocity that is 1 order of magnitude higher than values computed with the Eulerian–Lagrangian scheme. This may be related to the comparably high (non-physical) stress at the top of the upper layer. At the end of the simulation, the snowpack has settled almost twice as much with the layer-based scheme, which highlights the impact of this conceptual difference. This effect may result from an overestimation of velocity with layer-based schemes. Following our proposed method, the ice volume fraction is higher in the lower part of the snowpack and reaches higher values (Fig. 12d). Furthermore, the ice volume fraction at the top of the snowpack does not change during the simulation since there is no stress from overburden mass. In contrast, for the layer-based scheme ice volume fraction grows at this location (Fig. 12c). This is again due to the chosen stress condition at the top. Of course this discrepancy becomes smaller as we increase the number of layers, and this effect may reduce. However this slight offset in the stress condition will always be present and lead to uncertainties. In the proposed computational approach the spatial resolution of processes can be easily changed to assess its impact on snowpack evolution. In a future study, it might be interesting to quantitatively compare results against Jafari et al. (2020), who also rely on a rather fine spatial resolution.

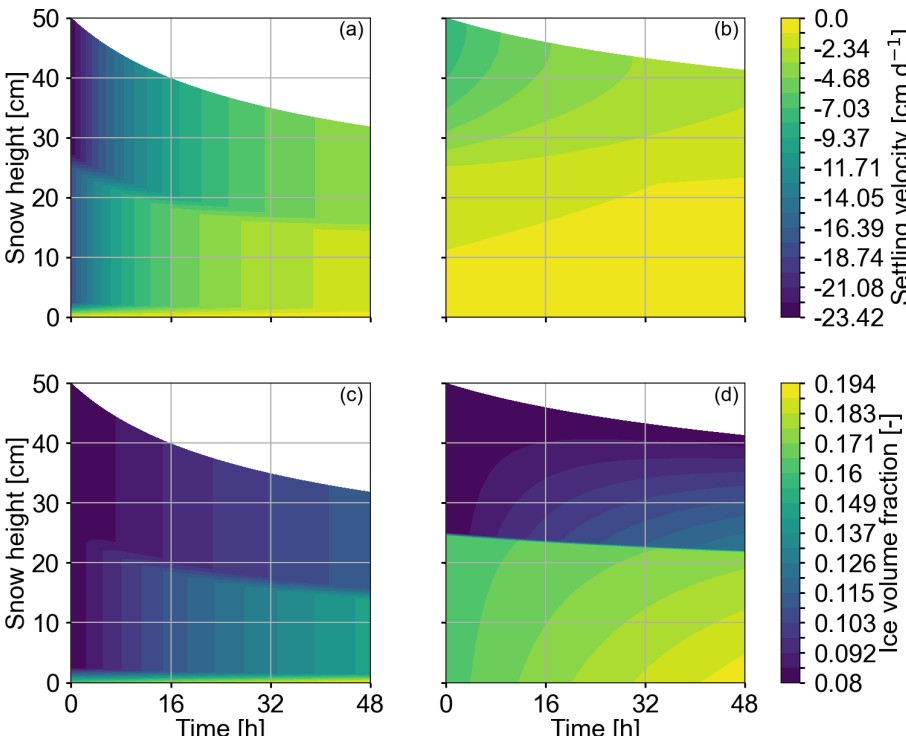

**Figure 12.** The plots show the temporal evolution of vertical velocity (top row) and the ice volume fraction (bottom row) for Case 6 of Table 2 corresponding to solely active mechanical settling. The *y* axis depicts snow height. For **(b)** and **(d)**, we applied our highly discretized settling scheme, and for **(a)** and **(c)**, we mimicked the layerwise discretization of layer-based schemes. Snow viscosity is controlled by the ice volume fraction alone, since heat transport is inactive. In **(a)** and **(c)** the lower and upper layer are resolved as the darker upper and brighter lower parts of the snowpack. Their respective values refer to the computational nodes at the top and between the two layers. The values retrieved for the lowest node do not represent an entire layer and are depicted at height zero. For the layer-based scheme, one velocity or ice volume fraction value represents the movement or density of the entire layer. In contrast, with our approach vertical velocity varies throughout each layer in **(b)** so that the ice volume fraction increases within layers and develops a gradual pattern **(d)**. For **(b)** and **(d)**, we interpret the locations of the upper and lower layers as the darker upper and brighter lower areas, respectively, in **(d)**.

## 4.8 Thin layers at the top and bottom of the snowpack (Case 5)

For the final scenario we implement a thin layer at the top and at the bottom of the two-layer snowpack. With this test case we want to show our model's feasibility for a potential future comparison with operational snowpack models such as Crocus that sustain layers at the top and bottom of the snowpack (Vionnet et al., 2012).

The initial condition for snow density is in principle equivalent to Fig. 3 except that the upper and lower 2 cm now form a new layer each, of 50 and 200 kg m$^{-3}$, respectively. The transition to the neighboring layer is linearly smoothed out over 1 cm for both thin layers. Figure 13a shows the ice volume fraction profiles for three times. The profile for 0 h reflects the initial condition for the ice volume fraction. The three layer transitions are discernible as steps in the profiles. Figure 13b depicts the deposition rate evolution. Sublimation is stronger at the layer transitions but shows a different evolution at all transitions. While the uppermost layer transition remains at a constant sublimation rate with time, sublimation

at the lowermost layer transition is very strong in the beginning and then decreases. The central layer transition shows that sublimation continuously increases until the end of the simulation time. This is consistent with our observations in Fig. 7b. Figure 7a and b show that after 48 h the lowermost thin layer has been reduced to less than half its initial height while the upper thin layer's thickness has remained almost constant. We suggest that the initially strong sublimation for the lowermost transition area is related to high temperature gradients at the lower boundary at the start of the simulation. Sublimation then reduces due to effects from consolidation. The effects of settling are very small in the vicinity of the uppermost layer transition, and the density difference in the layers is only 25 kg m$^{-3}$, which explains the constant and intermediate sublimation rate.

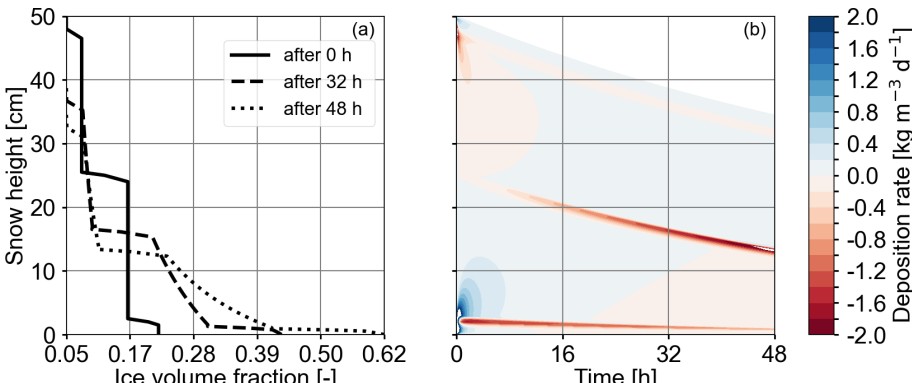

**Figure 13.** The simulations were carried out based on Case 5 of Table 2, which is the fully coupled processes with constant viscosity. Panel **(a)** shows the profiles for the ice volume fraction ($x$ axis) over snow height ($y$ axis) for a test case that has a thin layer at the top and bottom of the snowpack. The line corresponding to 0 h shows the initial condition for the ice volume fraction. The thin layers at the top and bottom each have a thickness of 2 cm. Panel **(b)** depicts the evolution of the deposition rate. The three transition areas of the four layers are characterized by increased sublimation rates (after 16 h at snow heights 42, 20 and 2 cm). The lobes at the top and bottom at the start of the simulation are due to the strong phase change activity triggered by the initial and boundary conditions.

## 5   Summary and conclusions

In this paper, we described in detail a hybrid Eulerian–Lagrangian computational approach to model the evolution of a dry snowpack. The model accounts for transport of heat and vapor, phase changes (sublimation and deposition), and mechanical settling processes. The ice mass balance is explicitly accounted for in the model formulation. It captures the evolution of the ice volume fraction in response to settling and phase changes. It constitutes an advection-dominated partial differential equation of hyperbolic type and is therefore conveniently solved with the method of characteristics, a popular Lagrangian-type scheme for such processes. Here, Lagrangian refers to the fact that the scheme tracks the motion of small reference volumes within the snow column by adjusting the node positions while at the same time accounting for phase changes within the moving snow. Solving the ice mass balance requires us to specify the vertical velocity as well as the mass production rate (sublimation rate/deposition rate). A closure for the velocity is derived by combining a common mechanical stress–strain relation with Glen's law and numerically approximating the resulting integrals. The deposition rate is due to vapor transport through a varying temperature field and can be determined from a diffusive-type process model that accounts for simultaneous heat and vapor transport. Due to its diffusive type (parabolic), a fixed-grid approach is most appropriate, referred to as an Eulerian approach. More specifically, we solved coupled heat and vapor transport by means of a first-order implicit-in-time finite-difference approximation. The Eulerian scheme for the process model's diffusive part complies with the non-uniform mesh that results from the Lagrangian scheme for the ice volume fraction evolution. In order to solve the complete dry-snow process model for the coupled evolution of the ice

volume fraction, temperature field, vapor field and settling processes, the Eulerian and Lagrangian parts are iteratively applied following a straightforward operator split approach.

We have implemented our proposed numerical scheme as a series of sequential updates within one simulation time step. The implementation follows a modular, extendable approach, such that each process building block can easily be considered or neglected for verification or validation purposes. We applied our numerical core to conduct a series of simulation scenarios comprising isolated processes (pure settling, pure heat transport), two-process coupling scenarios (heat transport in the presence of settling, coupled heat and vapor transport) and fully coupled processes (heat and vapor transport in the presence of settling). We furthermore investigated different viscosity closures as well as a linear and a non-linear version of Glen's law. A two-layer snowpack, consisting of a lower layer of higher density and an upper layer of lower density, served as a test case to demonstrate the feasibility of our approach. We simulated fields and profiles for the temperature, deposition rate, ice volume fraction and vertical velocity with a high spatial ($\sim$ mm to cm) and temporal ($\sim$ s to min) resolution.

We showed that our model implementation facilitates the comparison of various parametrizations and processes. This is enabled by the following:

– Our Eulerian–Lagrangian scheme along with its vectorized implementation is flexible and extendable. Alternative model closures, e.g., for the viscosity and the vertical velocity, can easily be integrated. To close for the velocity, we have successfully tested a non-linear strain rate closure commonly used in firn models (Lundin et al., 2017). The Lagrangian part of the solver (that accounts for the evolution of the ice volume fraction) can be singled out and coupled to an alternative process

model, e.g., when accounting for firn conditions instead of dry snow.

- The combination of an implicit Eulerian routine for the diffusion-dominated operators (that controls the time stepping) and a Lagrangian routine for the advection-dominated operators ran stably and robustly for all considered simulation cases (different viscosity closures, different versions of Glen's law).

- The numerical scheme allows for a high spatial resolution that resolves processes on the sublayer level. By construction, it relies on a mechanically consistent vertical velocity. This improves the accuracy since it makes the ad hoc specification of an artificial stress value for the uppermost layer (e.g., Vionnet et al., 2012), as required for conventional layer-based schemes, obsolete.

- The modular setup of the software allows a systematic study of various model formulations, in which we selectively considered different combinations of process building blocks without fine-tuning the stability of the solver. This is important to enable empirical–numerical investigations of the relevance of different process couplings.

- The incorporation of the higher-order mesh errors into the vapor and heat transport equations that account for deviations due to the non-uniform mesh increases accuracy especially in areas of large variations in node distance and of high temperature gradients.

In this paper, we mostly relied on numerical approximations that are of either first order (time integration and operator splitting) or second order (diffusion operator); the corresponding numerical solvers can be extended without conceptual difficulty, e.g., changing from a first-order time integration to a higher-order-in-time integrator. We commented on this in the relevant section (Sect. 3). Our simulation consistently showed that vapor transport and phase change in the presence of strong temperature gradients can induce a stronger phase change activity and in particular a localized sublimation rate peak above the transition area between two layers. We furthermore showed that this has the potential to result in a localized ice volume fraction reduction above the transition area. This in itself is not new, as a similar behavior has been deduced in Hansen and Foslien (2015) and analyzed in detail in our companion paper (Schürholt et al., 2021). In addition to the existing results, we have shown that the increased phase change activity persists in the presence of settling (even more pronounced), for a constant-viscosity closure in combination with a linear as well as a non-linear version of Glen's law.

## 6  Future work and challenges

In our paper, we deliberately focused on discussing modularity and extendability in the context of snowpack modeling, e.g., by assessing a whole process cascade for one relatively simplified physical setting. In order to discuss these aspects in depth, we restricted ourselves to one relatively simple physical setting. We are well aware that as of today, our proposed numerical approach is not ready for operational use, and that was not our intention. At this point in time, we rather would like to contribute to the discussion on how future snowpack modeling can benefit from a consistently formulated, hybrid Eulerian–Lagrangian solver. Nevertheless, it is important to discuss whether the suggested scheme is amenable to further extensions required for an operational snowpack model.

Most importantly, the proposed scheme would need a generalization for surface mass gain (precipitation) or losses (sublimation). This bears two technical challenges. First, concurrent settling and precipitation result in a non-monotonic vertical motion of the snowpack's upper surface, for which several techniques have been proposed in the past, e.g., based on a regularization approach (Wingham, 2000) or via kinetic boundary conditions as applied to sedimentation on ocean floors in Audet and Fowler (1992). A straightforward approach based on appending the computational grid sequentially during precipitation events likewise seems computationally feasible. A second challenge associated with the incorporation of precipitation events is the question of how to initialize the complete state (temperature, vapor and deposition rate) in the new snow layers. The latter is less critical in conventional layer-based schemes, as the necessary information reduces to "one value per layer". While the first challenge mostly consists in overcoming technical subtleties in the actual implementation, the second requires a thoughtful formulation of physically consistent boundary conditions. Neither of the two challenges seems to pose a severe risk.

Another important addition to our proposed snowpack model is the presence of liquid water in the snow. Conceptually, similar modeling approaches could be used to derive a model for wet snow. While including potential phase changes from melting and freezing could be straightforwardly implemented via the source term $c$, it is the advective transport of liquid water that is more demanding. Liquid water transport is commonly modeled via the Richards equation (Wever et al., 2014) which would benefit from existing hybrid Eulerian–Lagrangian solution strategies, as shown for saturated media without mechanical settling (Huang et al., 1994). Furthermore, a model for wet snow requires a second energy balance to account for the liquid water temperature. Once set up, it can be integrated into our computational workflow (Fig. 2).

Finally, operational models generally include the capability to account for topological change within the snow column, to capture either layer coalescence if two initially sep-

arated snow layers merge into one or layer separation if an initially homogeneous layer splits into two. By construction, our computational approach does not require a dedicated treatment for layer coalescence or separation. Both are implicitly accounted for in the continuous description of stratigraphy as long as the feature that is to be resolved is larger than the chosen spatial resolution of the computational grid. In the current version of our model we use a dynamic time step adaptation based on the mesh Fourier number, which leads to a decreasing time step size from 40 s at initiation to 3 s after 48 h. Layer coalescence, e.g., after certain time intervals or whenever a specific time step size is undercut, could facilitate longer simulations runs. Otherwise the resolution can be increased to avoid layer split-ups for regions with high gradients. Furthermore, our approach prevents the complete degeneration of layers as the ice volume fraction is constrained by the snow's maximum apparent density per construction of the scheme. Yet, while the theory suggests that layer coalescence and separation are not problematic, there might still be troublesome realistic test cases, especially when thinking about long simulation times. In order to address these and verify robustness, a series of benchmark tests have to be conducted. If necessary, the Eulerian–Lagrangian scheme in its current version can be equipped with occasional re-meshing (along with a re-sampling of field variables) triggered by the degeneration of well-defined mesh quality criteria.

We believe that a flexible and extendable computational approach, such as the one described in this paper, will be key for future snowpack modeling to facilitate the use of standardized benchmark problems (potentially used during a continuous integration) and allow us to systematically assess the model's predictive power, including uncertainty quantification, parameter estimation and model selection.

## Appendix A: Formulas required for the process model

### A1 Vapor saturation density

An empirical expression for the vapor saturation density $\rho_v^{eq}(T)$ in terms of temperature $T$ is formulated based on the empirical formulation for vapor saturation pressure from Libbrecht (1999) and reads

$$\rho_v^{eq}(T) = \frac{\exp\left(-\frac{T_{ref}}{T}\right)}{fT}\left(a_0 + a_1(T - T_m) + a_2(T - T_m)^2\right),$$
(A1)

with coefficients $a_0 = 3.6636 \times 10^{12}\,\mathrm{kg\,m^{-1}\,s^{-2}}$, $a_1 = -1.3086 \times 10^8\,\mathrm{kg\,m^{-1}\,s^{-2}\,K^{-1}}$, $a_2 = -3.3793 \times 10^6\,\mathrm{kg\,m^{-1}\,s^{-2}\,K^{-2}}$, $f = 461.31\,\mathrm{J\,kg^{-1}\,K^{-1}}$, $T_m = 273.15\,\mathrm{K}$ and $T_{ref} = 6150\,\mathrm{K}$. $f$ is the specific gas constant for water vapor. Note that division by $fT$ accounts for the conversion from pressure (Pa) (as used in Part 1) to density ($\mathrm{kg\,m^{-3}}$).

### A2 Model parameters in the transport and phase change equations

The effective vapor mass diffusion coefficient $D_{eff}(\phi_i)$ in terms of the ice volume fraction $\phi_i$ is taken from Calonne et al. (2014) but is extended by the heaviside function $\Theta$ to hinder vapor diffusion for ice volumes above two-thirds:

$$D_{eff}(\phi_i) = D_0\left(1 - \frac{3}{2}\phi_i\right)\Theta\left(\frac{2}{3} - \phi_i\right),$$
(A2)

with $D_0 = 2.036 \times 10^{-5}\,\mathrm{m^2\,s^{-1}}$ being the vapor diffusion constant in air.

The effective thermal conductivity $k_{eff}(\phi_i)$ in terms of the ice volume fraction $\phi_i$ is taken from Calonne et al. (2011) and reads

$$k_{eff}(\phi_i) = a_0 + a_1(\phi_i\rho_i) + a_2(\rho_i\phi_i)^2,$$
(A3)

with coefficients $a_0 = 0.024$, $a_1 = -1.23 \times 10^{-4}$ and $a_2 = 2.5 \times 10^{-6}$ and ice density $\rho_i$.

The effective heat capacity $(\rho C)_{eff}(\phi_i)$ in terms of the ice volume fraction $\phi_i$ is taken from Calonne et al. (2014) and Hansen and Foslien (2015) and reads

$$(\rho C)_{eff}(\phi_i) = \phi_i\rho_i C_i + (1 - \phi_i)\rho_a, C_a,$$
(A4)

with $C_i$ being ice heat capacity, $C_a$ air heat capacity, $\rho_i$ ice density and $\rho_a$ air density.

### A3 Constant viscosity for the two-layer case

#### A3.1 Linear Glen's law, $\eta_{const,m=1}$

We derived intermediate ice volume fraction $\phi_{i,const} = 0.1125$ and temperature $T_{const} = 263\,\mathrm{K}$ values from the initial condition of the two-layer case and insert them as constants into Eq. (4).

#### A3.2 Non-linear Glen's law $\eta_{const,m=3}$

Equation (4) does not hold for the Glen exponent $m = 3$; therefore we derive an adjusted constant viscosity $\eta_{const,m=3}$ via the constitutive equation (Eq. 3)

$$\dot{\epsilon}_{lit} = \frac{1}{\eta_{const,m=3}}\sigma_{max}^m,$$
(A5)

with $\dot{\epsilon}_{lit} \equiv 10^{-6}\,\mathrm{s^{-1}}$ being a strain rate value from the literature (Johnson, 2011) and $\sigma_{max} \equiv 547.71\,\mathrm{Pa}$ the maximum stress value obtained from the initial snow density profile of the two-layer case. Equation (A5) is then solved for the constant viscosity $\eta_{const,m=3}$.

#### A3.3 Restrict infinite ice volume growth

To hinder infinite ice volume growth, the constant viscosity $\eta_{const,m}$ is combined with a power law that yields exponential growth of viscosity for cells with $\phi_i > 0.95$:

$$PL(\phi_i) = \exp(pl1\phi_i - pl2) + 1,$$
(A6)

with $pl1 = 690$ and $pl2 = 650$. The constant viscosity is then multiplied with the power law ($\eta_{\text{const},m}\text{PL}(\phi_{\text{i}})$) so that computational nodes with $\phi_{\text{i}} > 0.95$ are assigned and viscosity grows exponentially. Note that for better readability the multiplication with the power law is omitted in the equations of this paper.

## Appendix B: Higher-order mesh errors to correct for non-uniform mesh

For the temperature equation (Eq. 22) the higher-order mesh error is

$$
E_{\text{T}}\left(T_{k+1}^{n+1}, T_{k-1}^{n+1}\right) = \frac{\Delta t^n 2\beta_{T,k}}{\left(\Delta z_k^n\right)^2 + \left(\Delta z_{k-1}^n\right)^2} \frac{\Delta z_k^n - \Delta z_{k-1}^n}{\Delta z_k^n + \Delta z_{k-1}^n}
$$
$$
\cdot \left(T_{k+1}^{n+1} + T_{k-1}^{n+1}\right), \tag{B1}
$$

and for the vapor transport equation (Eq. 23) it is

$$
E_{\text{c}}\left(T_{k+1}^{n+1}, T_{k-1}^{n+1}\right) = \frac{2\beta_{\text{c},k}}{\left(\Delta z_k^n\right)^2 + \left(\Delta z_{k-1}^n\right)^2} \frac{\Delta z_k^n - \Delta z_{k-1}^n}{\Delta z_k^n + \Delta z_{k-1}^n}
$$
$$
\cdot \left(T_{k+1}^{n+1} + T_{k-1}^{n+1}\right). \tag{B2}
$$

For the matrix equations Eqs. (25) and (24) the higher-order mesh errors are defined as $\mathbf{E}_{\text{T}}$ and $\mathbf{E}_{\text{c}}$.

## Appendix C: Matrices from temperature and vapor transport equations

Matrix $\mathbf{A}$ is defined as follows:

$$
\mathbf{A} = \begin{pmatrix}
A_{m,0}^n & 0 & 0 & \cdots & 0 & 0 & 0 \\
A_{l,1}^n & A_{m,1}^n & A_{u,1}^n & \cdots & 0 & 0 & 0 \\
0 & A_{l,2}^n & A_{m,2}^n & \cdots & 0 & 0 & 0 \\
\vdots & \vdots & \vdots & \ddots & \vdots & \vdots & \vdots \\
0 & 0 & 0 & \cdots & A_{m,nz-2}^n & A_{u,nz-2}^n & 0 \\
0 & 0 & 0 & \cdots & A_{l,nz-1}^n & A_{m,nz-1}^n & A_{u,nz-1}^n \\
0 & 0 & 0 & \cdots & 0 & 0 & A_{m,nz}^n
\end{pmatrix}. \tag{C1}
$$

For the heat equation (Eq. 24) ($\mathbf{A}_{\text{T}}$) the entries are

$$
A_{l,k}^n = \Delta t^n \left( \frac{\beta_{T,k+1}^n - \beta_{T,k-1}^n}{\left(\Delta z_k^n + \Delta z_k^{n+1}\right)^2} - D_{T,k}^n \right), \tag{C2}
$$

$$
A_{u,k}^n = -\Delta t^n \left( \frac{\beta_{T,k+1}^n - \beta_{T,k-1}^n}{\left(\Delta z_k^n + \Delta z_k^{n+1}\right)^2} + D_{T,k}^n \right), \tag{C3}
$$

$$
A_{m,k}^n = \alpha_{T,k}^n + 2\Delta t^n D_{T,k}^n, \text{ with } A_{m,0}^n = 1 \text{ and } A_{m,nz}^n = 1, \tag{C4}
$$

and for the vapor transport (Eq. 25) ($\mathbf{A}_c$) the entries are

$$
A_{l,k}^n = \frac{\beta_{k+1}^n - \beta_{k-1}^n}{(\Delta z_k^n + \Delta z_k^{n+1})^2} - D_{\text{c},k}^n, \tag{C5}
$$

$$
A_{u,k}^n = -\frac{\beta_{k+1}^n - \beta_{k-1}^n}{(\Delta z_k^n + \Delta z_k^{n+1})^2} - D_{\text{c},k}^n, \tag{C6}
$$

$$
A_{m,k}^n = \frac{\alpha_{\text{c},k}^n}{\Delta t^n} + 2D_{\text{c},k}^n, \text{ with } A_{m,0}^n = -\frac{\alpha_{\text{c},0}^n}{\Delta t^n} \text{ and } A_{m,nz}^n
$$
$$
= -\frac{\alpha_{\text{c},nz}^n}{\Delta t^n}, \tag{C7}
$$

with

$$
D_{f,k}^n = \frac{2\beta_{f,k}^n}{(\Delta z_k^n)^2 + (\Delta z_{k-1}^n)^2} \text{ for } f \in \{T, c\}. \tag{C8}
$$

Matrix $\mathbf{B}$ is defined as follows:

$$
\mathbf{B} = \begin{pmatrix}
\alpha_{m,0}^n & 0 & 0 & \cdots & 0 & 0 & 0 \\
0 & \alpha_{m,1}^n & 0 & \cdots & 0 & 0 & 0 \\
0 & 0 & \alpha_{m,2}^n & \cdots & 0 & 0 & 0 \\
\vdots & \vdots & \vdots & \ddots & \vdots & \vdots & \vdots \\
0 & 0 & 0 & \cdots & \alpha_{m,nz-2}^n & 0 & 0 \\
0 & 0 & 0 & \cdots & 0 & \alpha_{m,nz-1}^n & 0 \\
0 & 0 & 0 & \cdots & 0 & 0 & \alpha_{m,nz}^n
\end{pmatrix}. \tag{C9}
$$

For Eq. (24) ($\mathbf{B}_{\text{T}}$) the entries are

$$
\alpha_{m,k}^n = \alpha_{T,k}^n, \tag{C10}
$$

and for Eq. (25) ($\mathbf{B}_c$) they are

$$
\alpha_{m,k}^n = \frac{\alpha_{\text{c},k}^n}{\Delta t^n}. \tag{C11}
$$

Matrix $\mathbf{E}$ is defined as follows:

$$
\mathbf{E} = \begin{pmatrix}
0 & 0 & 0 & \cdots & 0 & 0 & 0 \\
E_{l,0}^n & 0 & E_{u,1}^n & \cdots & 0 & 0 & 0 \\
0 & E_{l,1}^n & 0 & \cdots & 0 & 0 & 0 \\
\vdots & \vdots & \vdots & \ddots & \vdots & \vdots & \vdots \\
0 & 0 & 0 & \cdots & 0 & E_{u,nz-2}^n & 0 \\
0 & 0 & 0 & \cdots & E_{l,nz-1}^n & 0 & E_{u,nz-1}^n \\
0 & 0 & 0 & \cdots & 0 & 0 & 0
\end{pmatrix}, \tag{C12}
$$

consisting of the terms for the heat equation ($\mathbf{E}_T$) (Eq. 24)

$$E_{l,k}^n = -\Delta t^n D_k^n \frac{\Delta z_k^n - \Delta z_{k-1}^n}{\Delta z_k^n + \Delta z_{k-1}^n} \text{ and} \tag{C13}$$

$$E_{u,k}^n = \Delta t^n D_k^n \frac{\Delta z_k^n - \Delta z_{k-1}^n}{\Delta z_k^n + \Delta z_{k-1}^n} \tag{C14}$$

and for the vapor equation ($\mathbf{E}_c$) (Eq. 25)

$$E_{l,k}^n = -D_k^n \frac{\Delta z_k^n - \Delta z_{k-1}^n}{\Delta z_k^n + \Delta z_{k-1}^n} \text{ and} \tag{C15}$$

$$E_{u,k}^n = D_k^n \frac{\Delta z_k^n - \Delta z_{k-1}^n}{\Delta z_k^n + \Delta z_{k-1}^n}. \tag{C16}$$

## Appendix D: Additional figures

### D1 Non-linear Glen's law

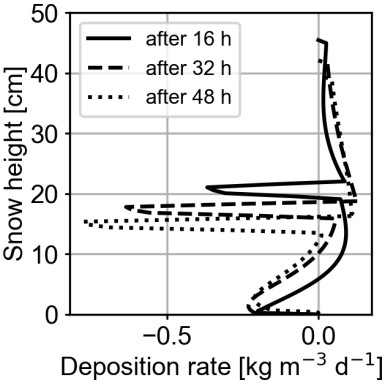

**Figure D1.** The plot shows the deposition rate profiles for 16, 32 and 48 h simulation times for Case 8 (Table 2), which is the fully coupled processes combined with the non-linear version of Glen's law. The $y$ axis depicts snow height.

*Code availability.* The model code is available via https://doi.org/10.5281/zenodo.5588308 (Simson and Kowalski, 2021).

*Author contributions.* The study concept, underlying model and methodology were devised by AS, HL and JK. Model analysis, software implementation and simulation runs were performed by AS supervised by JK. Test case analysis and discussion, data visualization, and manuscript preparation were carried out by AS with contributions from JK and HL.

*Competing interests.* The authors declare that they have no conflict of interest.

*Acknowledgements.* We thank the two anonymous reviewers for their helpful suggestions on the manuscript. The authors were supported by the Helmholtz School for Data Science in Life, Earth and Energy (HDS-LEE). The work was furthermore supported by the Federal Ministry of Economic Affairs and Energy, on the basis of a decision by the German Bundestag (50 NA 1908).

*Financial support.* This open-access publication was funded by the RWTH Aachen University.

*Review statement.* This paper was edited by Mark Flanner and reviewed by two anonymous referees.

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
