# Peer review of "Elements of future snowpack modeling - part 2: A modular and extendable Eulerian–Lagrangian numerical scheme for coupled transport, phase changes and settling processes"

_The Cryosphere, 2021_

## Author Comment (AC2)

**Cumulative response to Referee #1 and #2 tc-2021-73**

August 13, 2021

We thank both referees for their valuable comments and suggestions that significantly improve the quality of the manuscript. In this cumulative response we first reply to Referee#1 and then to Referee#2. To avoid redundancy, we refer to responses to the other referee whenever appropriate. Note, that we refer to three different versions of the manuscript in our replies: 1) the original version of the manuscript (equivalent to production file) to which the referees responded and to which all mentioned line numbers refer, 2) the revised manuscript already including many of the referee's suggestions that can be retraced in the attached latexdiff file generated with the original version, and 3) the final version of the manuscript, for which we propose changes, but that have not yet been implemented. It will rather be submitted and prepared after the editor's decision.

**Colour code:** Black: Referee comments purple: Author responses

**1 Referee #1 comments**

This paper describes a modular Eulerian-Langrangian approach solving the coupled non-linear processes. The authors report the shortcomings of the majority of snow models that do not give an explicit numerical solution of the ice mass continuity equation but are built around. The model formulation of the snow scheme presented explicitly accounts for the ice mass balance and couples mechanical settling, heat transport and vapour transport whereby these processes can also be analysed individually due to the modular structure of the model. This modular design of the new approach allows for high degree of flexibility, which the authors use to analyse different isolated and coupled scenarios involving of heat transport, vapour transport and mechanical settling of a dry two-layer snowpack. The work provides a good basis for future work and discussions on the future generation of snow models. The work is very exciting and gives new approaches that will be useful for the snow modelling community. However, I have some suggestions for changes in restructuring of the work and some visualisations.

**1.1 Major Comments:**

1. The discussion section lacks comparisons to other studies and observations from which one can see the progress. This would provide an important context for the presented results and an evaluation of the presented results of the different cases and model behaviour. More discussion and comparison would for example be desirable in section 4.2: are there other studies that show the same? Also section 4.3: How about a comparison to observations and other work

**e.g. for lines 408 to 410?**

We agree to the referee that a comparison of the implemented process model with data from lab experiments or field observations would be very valuable to the community and is a necessary next step in going forward. After intense discussion, we have decided to stick to the original scope of the manuscript, namely to discuss synthetic examples only. We still believe that our paper is of value-add to the research community due to the following reasons:

- A number of papers exist that compare process models with observations. Only few papers, however, compare the different process building blocks and corresponding explicit statement of their generic and numerical approaches. We deliberately focus on the synthetic model as it allows us to conduct a systematic qualitative and quantitative investigation of the coupling between the individual process building blocks. Including a comparison to data as the same level of detail would imply a significant extension to the existing manuscript.
- Our aim with this manuscript is to report on analytical findings to prepare the development of a new rigorous, generic snowpack model. Specifically, we see our progress as an extension of Hansen and Foslien (2015)'s model, namely the consideration of settling, which is one of the novelties in our work.
- While our computational model facilitates the above analysis, its current development stage does not allow for a meaningful comparison to experimental, or field data. This would require significant extensions to the code and the text, and is beyond the scope of this paper also in light of the 12 pages specification of TC that are already exceeded.
- In order to connect our synthetic results better with relevant physical regimes, we propose to a) elaborate at a greater level of detail on the model parameters used in this manuscript, for instance deposition rates or settling velocities, as this will help the reader to categorize simulated magnitudes, and b) we suggest to include an additional (yet also synthetic) numerical test case based on the suggestion of Referee#2 (line 420) with thin snow layers at the top and bottom of the snowpack. This test case points towards the feasibility of potential future comparative studies with detailed and more complex snowpack data.
- 2. The authors also use references sparingly in the method sections. For example, section 2.3 get along without using a single reference. If this is a result of the author's work, I recommend emphasising this to the reader.

Thanks for the remark. We assume that you refer to Sect. 2.2. because Sect. 2.3 has a number of references. We provide an extensive state of the art, and therefore decided to keep the text in the method's section slim. However, we agree that it is important to point out the sources that inspired our method's section. In the revised manuscript, we therefore included several references into the method's section as well.

3. I recommend restructuring section 4 and using a separate method section for the difference steps you took to apply the model. I suggest moving line 363 – line 379 to a subsection of the methods. I also recommend moving section 4.1 to the methods. For me, the results section starts with line 392.

Thanks for the suggestion. We included that in the revised manuscript and moved line 363 - line 379 to Section 3.6 **Application of the model** and Section 4.1. to Section 3.7 and made some small adjustments resulting from the structure changes in the text.

4. Section 4.6 and Fig. 10: Here you write that you compare cases 5 and 8 and use Fig. 10 for visualisation. However, Fig. 10 only visualise for case 8. I can only find the visualisation for

case 5 in Fig. 6 and 7 but only for different variables or different visualisation. This makes it difficult for the reader to compare the two cases. I suggest adding the visualisation for case 5 in Fig. 10. Also, I cannot find a comparison to case 5 in the text. You describe Fig. 10, briefly compare your results to section 4.2, which deals with case 1, and mention the sublimation peak as observed for other cases. However, I cannot find a comparison to case 8.

This is a good comment. Indeed no comparison to linear Glen's law was included in the original version of the manuscript o that the original subsection title was misleading. In the revised manuscript, we changed the subsection title to **Non-linear Glen's law in a fully coupled dry snowpack model of constant viscosity (Case 8)**. We decided to keep a brief comparative discussion of the settling profiles corresponding to linear versus non-linear Glen's law in the text of the revised manuscript. In doing so, we refer to the linear version of the settling profile for Case 1 in the previous Fig. 4 for comparison with Case 8. Although Case 1 in Fig. 4 is not fully coupled (in contrast to Case 8) the velocity profile can be used for meaningful comparison between the two.

**1.2** Detailed Comments:**

- Line 20: I suggest to add "e.g. Snowpack, Crocus" behind "when not even detailed snowpack models".
  - Done!
- Line 29: Please name the snowpack models. We named the snowpack models and added SNOWPACK and Crocus in the revised manuscript.
- Line 46: The vapour transport is very important and interesting. I encourage the authors to explore further the importance and the differences in e.g. alpine and arctic snow of the importance.

Thanks for the remark. In the revised manuscript, we elaborated further on the differences of vapor transport in alpine and arctic snow as follows:

Temperature gradients between ground and atmosphere imply upward vapor fluxes in snowpacks. Stronger temperature gradients (either due to a smaller snowpack height or colder surface temperature) in the Arctic yield higher vapor fluxes (Domine et al. 2019) compared with alpine snowpacks. Depth hoar layer with reduced density and thermal conductivity form at the snowpack's bottom. In alpine snowpacks similar hoar layers develop within the snowpack that may cause avalanches due to their low mechanical stability (Schweizer et al. 2003).

- Line 50-52: See comment above: Please describe in more detail why it is important. This is answered in the previous comment for line 46.
- Line 57: Please also add the details for Crocus.
  We agree with this comment. However, we could not find a discussion on explicit time step constrains in the literature. We found possible time step sizes of 15 min in Viallon-Galinier et al. (2020); Brun et al. (1989) and 1 hour in combination with other models in Vionnet et al. (2012) and added these details to the revised manuscript.
- Line 61: You mention that a finer spatial resolution is needed. Please mention the spatial resolution of SNOWPACK and Crocus. We agree with the referee that this information would be valuable. We added the possible values for layer thicknesses based on a literature study to the revised manuscript. We found that the minimum layer thickness in Crocus is 0.5 cm (Brun et al., 1989). An upper limit for layer thickness is not stated in the referenced literature. In

SNOWPACK, a typical layer thickness is 2 cm (Wever et al. 2016) but can also reduce to 0.01 cm Jafari et al. (2020). Explicit ranges that constrain layer thicknesses in SNOWPACK and Crocus are not given in the referenced literature.

- Line 74: Reference is needed for the current treatment of densification in snowpack models. We added references to the revised manuscript.
- Line 80: Define " $\sigma$ -coordinates".  $\sigma$ -coordinates are used in a number of disciplines, yet are particularly well known from oceanography: The ocean's surface and bottom are projected on coordinates  $\sigma = 0$  and  $\sigma = -1$  that follow the ocean floor's topography. We included a definition in the revised manuscript.
- Line 111: Please add a reference where this common starting point is used in snow models. We added references and an explanation to the revised manuscript.
- Line 113: I recommend deleting "without explicitly mentioning every time". You already write "if not stated otherwise". We followed the referee's suggestion and deleted it in the revised manuscript.
- Line 115: "snow density" did you mean "ice density" ? Please clarify. Thanks for the remark. We meant snow density. We reformulated this passage in the revised manuscript for clarification.
- Line 114-117: Reference is needed here. We added references to the revised manuscript.
- Line 119: Would there be a lot change if wet snow were used? Please clarify in 2-3 sentences the differences that dry snow and wet snow would make (more detailed description of what you have already started in line 121).

Thanks for the remark. In the original manuscript, we discuss how to incorporate water in the summary and conclusions section. In response to your comment, we referenced in line 122 the summary and conclusions section for details, and we extended the corresponding discussion in the revised manuscript.

- Line 121: I recommend adding "compared to wet snow situation" after "as the more challenging (yet less investigated) one" We added the referee's suggestion.
- Line 122: I suggest starting a new paragraph for "Note, that water transport ..." We adjusted the paragraph.
- Table 1: Is there a value missing for density? I suggest ordering the variables within the headings according to their occurrence in the equations. So ice volume fraction first, followed by vertical velocity, etc. This is a helpful suggestion. We changed the table accordingly in the revised manuscript. Air

density is now included. Furthermore, we noticed that  $k_i$  and  $k_a$  are not used in the paper and deleted them, while stress related parameters are not listed yet and included them. We changed the name "Effective model parameters of snow" to "Model parameters of snow". We changed the term  $\rho_{eff}$  to  $\rho_{snow}$  (consistently in the whole manuscript).

• Line 129: "source term c" do you mean ice deposition rate? Thanks for the remark! Indeed, this can be understood as the ice deposition rate. However, our intention is to keep c more general since it can also refer to more processes such as ice production/loss by freezing/melting additional to sublimation/deposition. We therefore decided to keep the general definition of c in the beginning of the methods sections, also due to the comment on line 132-134 that asks for the impact of melting on the continuity response. We also added an additional clarifying explanation in the revised manuscript.

• Line 132 – 134: You write that vertical motion results either from mechanical settling changes in ice volume from sublimation/deposition. What about melting and snow redistribution? We added melting and freezing as contributors to the source term and the continuity response in the revised manuscript. Snow redistribution in form of compaction, heat and vapor transport as well as phase changes is included in the vertical motion.

• Line 139:

see comment for line 129 See our reply to comments for line 129 and line 132-134.

- Line 141: "Hence we will refer to c as the deposition rate" I suggest to introduce this term earlier maybe line 129 (see comments line 129, line 139) See our reply to comments for line 129 and line 132-134.
- Line 152: Please add references for snowpacks where the approach typically chosen in the snowpack models is apparent. This is a good comment. Thanks for that! Even though this stress-strain relation is not explicitly stated in the literature of detailed snowpack models, these models implicitly fall back on this standardized approach. We added references, and for clarification we replaced 'typically' by 'implicitly' in the revised manuscript.
- Line 170: Which properties of the snow microstructure? We added the properties to the revised manuscript.
- Line 185f: Define g (only defined in line 299). We included the definition for g and also deleted 'Here,' in the revised manuscript.
- Line 186: Please define the term "snowpack's effective density" Thanks for the remark. We rather call it snow density, which is more intuitive, and changed it at all locations including Table 1 in the revised manuscript.
- Line 189: Please define ζ.
   We defined ζ as the integration variable in the revised manuscript.
- Line 203 210: First, you write that you are extending the model for mechanics but a few lines later you write "and thus it is neglected in the following." I cannot follow this thought. Please clarify This is a valuable comment, thanks! We agree that this was not clear in the original version of the manuscript. Our approach is as follows: We extended the model by Hansen and Foslien (2015) for mechanics. The convective term in the vapor equation arises from the mechanics. However, its impact on deposition rate, temperature, etc. is very low compared to that of diffusion. Thus, we neglect it in the remainder and essentially use the model by Hansen and Foslien (2015). To avoid any confusion, we deleted the sentences that discuss the mechanics extension and deleted the convective term from the vapor mass balance in the revised manuscript. This does not affect any of our results and conclusions!

- Line 214 215: "Instead of following [...] closure for the source term" repetition of line 202. I suggest to add the references mentioned in line 215 to line 202 and delete 214-215. Nice suggestion! We included this comment in the revised manuscript.
- Line 244: What do you mean with "necessary accuracy"? Please add a sentence what the necessary accuracy for the scenarios is. Thanks for the remark! We agree that 'necessary accuracy' does not make sense in this context. We replaced 'results at the necessary accuracy' by 'is applicable to the' in the revised manuscript.
- Line 245: I suggest to add "using a 1d snow column" at the end of the sentence "[...] scenarios considered in the paper." We incorporated this comment in the revised manuscript.
- Line 247:"In that situation". Which situation do you mean with "that"? What is a complex geometry in this context? 2d/3d? Thanks for the remark. What we mean with 'that situation' is a finite element solution. With complex geometry, we mean snowpacks that, for instance lay on a mountain slope. We reformulated the sentence in the revised manuscript to make that clearer.
- Line 252: For the interested reader, it would be helpful to add an example reference for the use of a second order Strang splitting. We added a reference to LeVeque (2002) to the revised manuscript, in which the splitting is described.
- Line 275: You only explain the usage of a Lagrangian approach in this line but already use the term before in line 237. I recommend moving this explanation to an earlier line. Thanks for the comment! We added an explanation to **Step 1** in the same style as for **Step 2** in the revised manuscript. We did not change the content of line 275, as we believe it increases readability.
- Line 301: What about Crocus? Please add reference here. We agree that it would be nice to include this information regarding Crocus. Yet, it is neither stated explicitly in the Crocus references nor did we find it after screening the source code.
- Line 313: "within" Corrected!
- Line 323: Define  $\alpha$ .

 $\alpha$  fulfills the role of a generalized density times heat capacity ( $\rho c_p$ ). It is used to formulate a generic numerical scheme, which is the aim of our work. Note, that we added the right hand side of the generalized equation (Eq. 21), which was missing in the original manuscript.

• Figure 2: Nice and helpful overview. I suggest inserting the names of Calonne, and Hansen and Folien in the boxes of the approaches and the equation numbers used to give the reader a quick overview.

This is a good suggestion. We will change the figure accordingly in the final version of the manuscript.

- Line 347: Figure appears in manuscript before it is mentioned in the paper. We will pay attention to that when preparing the final version of the manuscript.
- Line 349: What is the minimum time step for the output? Does it also vary? Thank you for this comment. In our approach, we chose a dynamic time step adaptation

based on the mesh Fourier number. In response to settling processes, the mesh sizes in our test cases vary and decrease, and so does the time step. At initiation the time step is in the range of 1 min. After 2 days simulation time the time step decreases to a few seconds and after 4 days simulation time to thirds of seconds. In our test cases, the time step size is mostly dominated by the mesh size. In order to facilitate longer simulations runs, one could think of merging homogeneous neighboring cells either after regular time instants, e.g. after a specific time period (day), or whenever a certain time step threshold is undercut. However, this is future work, and we did not explore it in the current study. We added some sentences on the decreasing time step size, and we corrected the equation by adding the subscript k to the formula in the revised manuscript. We will include a short discussion on the time step evolution in the summary and conclusions section of the final manuscript.

- Line 351 356: Already in Figure 2. Please bring both information together in Figure 2 (see also comment about Figure 2). We agree to this comment and reorganized text and caption in the revised manuscript. Additionally, we moved the last three sentences of the caption of Fig. 2 to the text (following Referee#2's comment on Figure 2).
- Line 360: I recommend deleting "(Sect. 4)" We deleted (Sect. 4) in the revised manuscript.
- Line 383: "The densities are in the range of [...]" For which regions/ type of snow is this the case?

Thanks for the remark. Here, we are not inspired by snow in a particular geographic region, but wanted to present an extreme and very active snow regime. The layered snowpack with small snow densities ensure strong dynamical coupling of the processes. We will include a discussion of geographical locations, where these snow densities/snowpacks can be found in the final version of the manuscript.

- Line 384: Correct "over over". Corrected!.
- Figure 3: Why do you choose these densities? Why does your layer have equal thickness? Representative for what region? Why do you use the values for ice volume, representative for what? Any reference for the used values? See the response to your previous comment on line 383.
- Line 392: I recommend changing "As the first step" to "first" We changed it in the revised manuscript.
- Line 394: "Furthermore, the vertical velocity varies less in the upper layer than it does in the lower layer." Misleading formulation, difficult to understand on first reading. Please clarify: in comparison of the same time step or between/within time steps? Thank you for this comment. We meant within one time step. We reformulated the sentence in the revised manuscript.
- Line 397: Does it increase at all in the upper layer? I can only see the thinning of the snowpack but no colour changes in the upper layer. Ice volume fraction also increases in the upper layer. However, the increase is small and not easily seen. We will adjust the color bar levels to improve visibility in the final version of the paper.

• Line 398: "the extent of the upper layer decreased only slightly with time", how many cm at the end?

It decreases by approximately 3.5 cm. We included this information in the revised manuscript.

- Figure 4: It might help the reader if you mark the upper and lower layers at least on the y-axis, as you write about the upper layer and the lower layer e.g. line 394, line 397. Thank your for the comment. We agree that this can be helpful. Instead of marking upper and lower layer in the result plots, we replaced Layer 1 by lower layer and Layer 2 by upper layer in Fig. 3 and changed the names at all relevant locations in the text of the revised manuscript. Furthermore, we improved the figure caption of Fig. 4 to clarify our interpretation of the locations of upper and lower layers in the revised manuscript. In the final manuscript, we will carry over this change to the rest of the figures.
- Line 420: You mention that this is explained in detail in Schürholt et al. but a short summary would be helpful. 1 to 2 sentences e.g. why, also in reality? Where observed? Why peak, why peak between layers?

This is clearly an important point! Although some explanations of the sublimation rate peak were already included further down in the section text, we agree that this can be emphasized more: In the revised manuscript, we extended the caption of Fig. 6 and the text to explain that the small peak in deposition rate of (a) (so to the right) is interpreted as the onset of the spatio-temporal oscillations. We also included a short reference to the results of Schürholt et al. (2021) in the revised manuscript.

- Line 421 426: Very interesting. It would be valuable if you could discuss this further and in relation to reality. What is expected in reality? Why is the peak 4 times higher? We agree that this is interesting. In order to focus more on this discussion, we reorganized the section and included an explanation why sublimation is stronger for the fully coupled processes. Results from an ongoing literature search regarding ranges observed in experiments will be includes it in the final version of the manuscript.
- Line 430: In equation 23 und 24, Ec and Et are used, but only one number is given here. The meaning of this is unclear. Shouldn't there be one number for Ec and one for Et? You are right, in principal both could be evaluated independently. However, this does not yield additional information, as the deposition rate is directly derived from temperature via the vapor transport equation. Still it might be interesting to look at both due to their non-linear relation.

We replaced 'error term' with 'higher-order mesh errors', as this is the focus of our error analysis. 'error' refers to a quantitative value, whereas the higher-order mesh error terms are a mathematical expression. We determine the error by computing the temperature deviation between the solution that considers higher order mesh error terms, and the solution that doesn't (quantified in an L1 norm). We reformulated the explanation in the revised manuscript. We also added a unit to the error K.

• Figure 6: It would be valuable for the work if you also add a plot c) for the temperature gradient for case 5. Perhaps a plot d) for vapour density gradient for case 5 over time would also be beneficial. You write about both in your text.

Thanks for this comment. These additional plots can easily be generated for the final version of the manuscript.

- Figure 7: (a) Where is the dashed line? If it coincides with the solid line, please change the visualisation so that both lines are visible. Indeed, the lines coincide. We will improve their visibility in the final version of the manuscript.
- Figure 8: Not mentioned in the text. Thanks for the comment. Indeed, a reference to the figure was missing in the text. We included a reference in the new subsection Heat and vapor transport coupled to settling with a dynamic viscosity (Case 4, 6, and 7) of the revised manuscript (see also response to your major comments on Section 4.6 and Fig. 10). Note that this also led us to change the order of Fig. 8 and 9.
- Line 441: I recommend adding at the end of the sentence "for case 7 of Table 2, which refers to the fully coupled process in combination with dynamically varying viscosity". We included the reference to Table 2 as suggested.
- Line 443 444: Please add a discussion why this is the case. Thanks for the remark. We added a short discussion.
- Line 449: "As discussed before" please add section number. Done!
- Line 452: I suggest adding "Figure 10 (a) [...] for case 8". We included this comment.
- Line 458: "decreases in height". I recommend providing numbers from what height to what height or by how many cm? It decreased by 9 cm. We added this detail to the revised manuscript.
- Line 456: You use alternately "paper, study and article" in your manuscript. I recommend deciding on one term for the whole manuscript. We changed it to paper at all relevant locations in the revised manuscript.
- Line 531: "in the respective section" Please add section number. We added the section reference in the revised manuscript.
- Line 537: (e.g. Vionnet et al. (2012) We added brackets to the reference in the revised manuscript.
- Section 5: In your summary and conclusion section, you use about one page to write about future work and challenges. I recommend moving this part to a separate section called e.g "Future work and challenges" before the summary and conclusion section. Thanks for this suggestion. We will reorganize Section 5 and split it into Future work and challenges and Summary and conclusions in the final version of the manuscript.
- Line 561: "Audet and Fowler, 192 please remove brackets. We removed the brackets accordingly in the revise manuscript.
- Line 607: Define  $\rho_a$ . We included a definition for  $\rho_a$  in the revised manuscript, and we furthermore added it to Table 1 (see reply to Table 1).

- Figure D1: Why are different times shown here than in the other plots (15h, 32 h, 48h vs. 0 h, 16 h, 48h)? I suggest making the graphic square. We will adjust the times of this plot according to the previous plots in the final version of the manuscript.
- Line 729: 2021. Thanks for spotting. We corrected the year of the reference in the revised manuscript.

**2 Referee #2 comments**

comment of Referee #1.

Simson et al. present rigorous development of a numerical model for coupled heat transport, vapour diffusion and settling in snow. I expected this to be a difficult paper to read, but I was pleasantly surprised by how readable and understandable it was. Because this is presented as a contribution towards snow model development rather than a full snow model, the test cases that can be considered are necessarily limited, but it is still disappointing that the paper contains no comparisons with observations at all.

**2.1 Specific comments listed by Line number**

- Line 2: The majority of models use non-deforming layers with ice and water moving between them and enforced conservation of mass. What we meant is the majority of detailed snowpack models. We clarified it in the text of the revised manuscript.
- Line 18: Doesn't the focus on snow water equivalent suggest that mass is the most important prognostic variable? The snow water equivalent is only one of many examples that shows the significance of the snow density.
- Line 23 Bartelt and Lehning (2002) do use a Lagrangian coordinate system that moves with the ice matrix, but "Lagrangian coordinate system that moves with the ice matrix" is not actually a quote from that paper.

We double checked the quote. It is a quote of the paper by Bartelt and Lehning (2002) (doi:10.1016/S0165-232X(02)00074-5) found on page 127 left column: " A Lagrangian coordinate system that moves with the ice matrix is employed. "

- Line 57 Considering this motivation from Domine et al., demonstrating whether modelled vapour transport can produce the sort of density stratification observed in shallow Arctic snowpacks subject to high temperature gradients would be an important test case. Thank you for this suggestion that we take under consideration for a potential comparative study with observational data in the future. Please also see our response to the first major
- Table 1 c and v are not state variables in the thermodynamic system sense; they can be derived from known temperature, ice volume fraction and vapour density. Excessive precision for latent heat of sublimation

We agree to this comment and corrected the table by moving c and v to the **Model parameters of snow** section in the revised manuscript. Please note, that further changes of Table 1 are due to Referee#1's comment on Table 1.

The constant values are due to Calonne et al. (2014). We converted the latent heat of sublimation from  $J/m^3$  to J/kg. As our computations are based on this latent heat value, we decided to keep the latent heat value in the table but rounded up the decimal.

- Line 129 Could note that the settling velocity was neglected in Part 1, equation 7 This is a good suggestion that we included in the revised manuscript.
- Line 216: Better to write vapour density with the eq subscript hereafter. We agree with the referee and replaced  $\rho_v$  by  $\rho_v^{eq}$  in the revised manuscript.
- Line 220: Equation 10 differs from the corresponding equation 5 in Part 1. Checking units, the error is actually in Part 1. Thank your for spotting. We will correct this error in Part 1.
- Figure 2 caption: The last three sentences don't really fit in a figure caption. We agree to this comment and moved the last three sentences to the text of the revised manuscript.
- Table 2 caption: Case 8 is also fully coupled. Thanks for the comment! We added Case 8 as fully coupled in the table caption of the revised manuscript.
- Figure 3: Why do the ice volume fraction and temperature axes run right to left? Thank you for the comment. We agree that direction of the axes does not make sense and changed them to left to right running axes in the revised manuscript. We also changed  $T_0$  to  $T^0$ .
- Figure 4: An additional plot with vertical profiles of density at 0, 16 and 48 hours could be interesting. Compaction from 150 to 420 kg/m3 over 2 days at the base of a 50 cm snowpack seems high if there were to be a comparison with observations (but not as implausible as Figure 10).

Levels on plot (b) appear quantized but the colour bar is not.

We agree with the referee that these plot would be a value-add. These additional plots can easily be generated for the final version of the manuscript. We will also adjust colour levels in (b) for the final version of the manuscript. Regarding the comment on observations please see our reply to Referee #1's for major comment 1. and line 383.

- Figure 7 Deposition rate would be better shown with a diverging colour scale centred on 0. This is a good suggestions, thanks! We assume that the referee meant Fig. 6 because Fig. 7 does not have a colour plot, and it does not show deposition rate. We will adjust all color scales for deposition rate to a diverging color scale in the final version of the manuscript.
- Figure 8: Reference to Figure 8 is missing in the text. Thanks for the remark. See our reply to to Referee#1 on Fig. 8..
- Line 470: A layer-based snowpack model could be viewed as having computational nodes in the centres of the layers. Overburden on the top layer being half the layer's weight then makes more sense. If using Vionnet et al. (2012) as an example, this two-layer, 50 cm snowpack is something that would not arise in Crocus; thin layers are maintained at the top and bottom of the snowpack for heat conduction calculations.

Thank you for this comment. In response to this comment, we suggest to include an additional synthetic test case with thin layers at the top and the bottom of the snowpack to comply

with a more realistic comparison to Crocus. This suggestion is also stated in our reply to major comment 1. of Referee#1. The way our model is setup, we do not intend to place computational nodes at the centre of the layers since the nodes rather constrain the extent of the individual numerical layers.

• Line 594: a0, a1, a2 and f all have units, which should be given. f is the gas constant for water vapour. This same formula was attributed to Mason (1971) in Part 1. Thanks for the remark. We added the units.

Minor corrections: Thanks for spotting! We included all of the following minor corrections in the revised manuscript.

- Line 25: "has been well established"
- Line 56:

"Both require"

- Line 57: "uses time steps on the order of 15 minutes or longer"
- Line 74: "do not take full advantage"
- Line 157: "depends on both the physical regime and computational feasibility"
- Line 161: "challenging to determine from experiments"
- Line 197: "In the remainder of this paper"
- Line 244: "results in the necessary accuracy"
- Line 253: "implemented in Python"
- Line 254: "but also allows"
- Line 305: "avoids numerically approximating"
- Line 309: The lengthy parenthetical clause "see for instance Sect. 3.4. in Bartelt and Lehning (2002) or its recent extension Jafari et al. (2020)" would be better inside parentheses than between commas.
- Line 317: "allows inferring the most plausible process model ... given certain data"
- Line 319: "results in a mesh"
- Line 530: "without conceptual difficulty"
- Line 568: "While including potential phase changes"
- Line 598: Part 1 used capital Theta for the Heaviside function.
- Line Appendix C: Matrix elements should be enclosed in brackets.

**3 Additional typo correction based on our own review**

For completeness, we also list minor typo-like changes that we implemented in the revised version of the manuscript. Note, that none of these changes affect the results.

- line 297: We changed the definition of the cell sizes  $\Delta z_j$  to  $\Delta z_j := z_{j+1} z_j$  with  $j \in [0, nz[$  to respect that the number of cells nc is always the number of nodes nz minus 1.
- Eq. (13), (14): minus instead of plus in front of  $\partial_z v \phi_i$
- Eq. (19): k instead of l in the sum index
- Eq. (21): derivative of temperature w.r.t. time instead of temperature
- Eq. (22): added latent heat L to  $\alpha_T$  and  $k_{eff}$  to  $\beta_T$
- Eqs. (23), (B1), (B2): added superscript n to  $\Delta z_{k-1}^n$
- Eq. (C8): added subscript c to  $\alpha_{c,k}$
- Eqs. (C11), (C8): Division by  $\Delta t^n$ .

**References**

[revised manuscript text omitted]

(5)

Here,  $\rho_{eff}$  g is the gravitational acceleration and  $\rho_{spow}$  refers to the snowpack's effective snow's density, which is clearly dominated by the ice fraction via  $p_{eff} \approx \phi_i(z) \rho_i \rho_{snow} \approx \phi_i(z) \rho_i$ . It varies with the position z in the snow column due to a 205 vertically varying ice volume fraction  $\phi_i(z)$ . Integration of Eq. (5) and combination with Eqs. (2) and (3) yields an expression for the velocity gradient:

$$\partial_z v = \frac{1}{\eta} \left( g \int_z^{H(t)} \phi_i(\zeta) \rho_i d\zeta \right)^m.$$
(6)

 $\zeta$  is the integration variable. A second integration along the vertical axis finally yields an expression for the velocity at position z in the snow column: 210

$$v(z) = -\int_{0}^{z} \frac{1}{\eta} \left( g \int_{\tilde{z}}^{H(t)} \phi_{i}(\zeta) \rho_{i} d\zeta \right)^{m} d\tilde{z},$$
(7)

in terms of total height H(t), ice volume fraction  $\phi_i(z,t)$ , and with  $v(z=0,t) \equiv 0$ . This definition of the vertical velocity yields a process that complies with the obvious physical constraints: a) the velocity vanishes at the bottom of the snow column, hence prevents artificial penetration into the ground. This is similar to displacement requirements in SNOWPACK (Bartelt and

Lehning, 2002). b) the vertical velocity accumulates with height, which prevents any artificial disaggregation of the snowpack, 215 and c) the vertical velocity relaxes towards zero as the ice volume fraction tends towards its maximum volume fraction  $\phi_i < \phi_i$  $\phi_{i,max} < 1$ . In the ongoing remainder of this paper, we will use Eq. (7) to account for the mechanical settling of the snowpack.

**2.4 Transport and phase changes**

The ice deposition rate c as relevant to solve Eq. (1) typically depends on a cascade of coupled heat and mass transport for the involved phases ice, water and vapor. In this article, we will consider a process model proposed by Hansen and Foslien (2015) 220 that reflects a dry snow condition in which void space is filled by vapor only. Note, however, that this coupled process model could readily be substituted or extended by another one, e.g. the one from Calonne et al. (2014), from Calonne et al. (2014), Jafari et al. (2020) or Schürholt et al. (20202021).

Next, we state the essential aspects and process equations of the model proposed in Hansen and Foslien (2015). We extend the model for mechanics and describe how it can be used to recover the ice deposition rate: 225

Assuming dry snow conditions, the ice production is solely determined by mass transport between vapor and ice. The vapor mass balance reads

$$\partial_t \left( \rho_v \left( 1 - \phi_i \right) \right) - \nabla \cdot \left( D_{eff} \nabla \rho_v \right) \underline{+} \underline{\rho_v \nabla \cdot v \phi_i} = -c, \tag{8}$$

in which  $\rho_v$  denotes the vapor density and  $D_{eff}$  the effective vapor diffusion coefficient. Note that the convective term on the

- 230 left hand side derives from mechanical settling and is thus not included in Hansen and Foslien (2015). Its effect on deposition rate is rather low, and thus it is neglected in the following. Vapor production corresponds to negative ice deposition rate -cthat represents sublimation. Following Hansen and Foslien (2015), vapor density in the pore space can be assumed to be at saturation density  $\rho_v^{eq}$ , so that  $\rho_v \equiv \rho_v^{eq}$ . The latter is well investigated, and empirical relations exist that specify its temperature dependency  $\rho_v^{eq}(T)$ . In this work, we will employ an empirical relation from Libbrecht (1999). The full expression can be read
- in Appendix A1. Instead of following the approach of Hansen and Foslien (2015) one could also use another closure for the source term (Calonne et al., 2014; Jafari et al., 2020; ?). Due to the closure for vapor density  $p_v \rho_{u,v}^{eq}$ , the vapor mass balance (Eq. (8)) can be rewritten using the temperature dependence of the equilibrium vapor density

$$(1-\phi_i)\frac{d\rho_v}{dT}\frac{d\rho_v^{eq}}{dT}\partial_t T - \nabla \cdot \left(D_{eff}\frac{d\rho_v}{dT}\frac{d\rho_v^{eq}}{dT}\nabla T\right) = -c.$$
(9)

Assuming the snow to be in thermal equilibrium at the microscale, we can likewise write the energy balance in terms of the temperature, which reads

$$(\rho C)_{eff} \partial_t T - \nabla \cdot (k_{eff} \nabla T) = cL.$$
(10)

The parameters  $(\rho C)_{eff}$  and  $k_{eff}$  stand for the effective heat capacity of snow and effective thermal conductivity, respectively. Both parameters depend on the ice volume fraction, and their definition is stated in Appendix A2. The right hand side of the heat equation accounts for latent heat release, which is coupled to phase change processes.

The system of the two equations, Eqs. (9) and (10), and the two unknowns, temperature T and deposition rate c, is solved by replacing c in Eq. (10) with Eq. (9), which yields a non-linear equation for temperature

$$\left(\left(\rho C\right)_{eff} + \left(1 - \phi_i\right) \frac{d\rho_v(T)}{dT} \frac{d\rho_v^{eq}(T)}{\sqrt{
[revised manuscript text omitted]
}_{\mathbf{T}}^{-1} \left( \mathbf{B}_{\mathbf{T}} \boldsymbol{T}^{n} + \underline{\mathbf{E}}_{\underline{T}} \underline{\mathbf{E}}_{\underline{T}} \boldsymbol{T}^{n} \right)$$

$$(25)$$

$$\boldsymbol{T}^{n+1} = \mathbf{A}_{\mathbf{T}}^{n+1} + \mathbf{D}_{\mathbf{T}} \boldsymbol{T}^{n} + \mathbf{D}_{\mathbf{T}} \boldsymbol{T}^{n}$$

$$(26)$$

 $\boldsymbol{c}^{n+1} = \mathbf{A}_{\mathbf{c}} \boldsymbol{T}^{n+1} + \mathbf{B}_{\mathbf{c}} \boldsymbol{T}^{n} + \underbrace{\mathbf{E}_{C}}_{\sim \sim} \underbrace{\mathbf{E}_{c}}_{\sim \sim} \boldsymbol{T}^{n}.$ (26)

First, Eq. (25) is solved for temperature  $T^{n+1}$ . Next, the updated temperature is used to solve Eq. (26) for the deposition rate  $c^{n+1}$ . The complete matrix definitions are given in Appendix C. Note, that formally, it would be possible to add up matrices  $\mathbf{B}_{T}$  and  $\mathbf{E}_{T}$  as well as  $\mathbf{B}_{c}$  and  $\mathbf{E}_{c}$ . We decided to keep them in this particular form to stress the similarity of this formulation with a standard finite difference approximation on an equidistant mesh, in which we are left with  $\mathbf{B}_{T}$  and  $\mathbf{E}_{c}$  and  $\mathbf{E}_{T}$  and  $\mathbf{E}_{c}$  vanish.

**3.5 Iterative coupling of Eulerian and Lagrangian solutions**

- 375 The derived numerical update routines for temperature, deposition rate, vertical velocity and ice volume fraction comprise the four main modules that are sequentially called to update the respective state variables for one time step. A schematic illustration is given in Fig. 2. The equations for heat and vapor transport have already been implemented by Calonne et al. (2014) and Hansen and Foslien (2015). A feedback on the ice volume fraction in the absence of a vertical velocity has been investigated in Part 1 of the companion paper (Schürholt et al., 20202021). The modules for vertical velocity and the coupled update of ice
- 380 volume fraction and mesh coordinates, through the method of characteristics is novel in our approach. Our implementation is modular in the sense that it allows for a coupling with other process models that comply with a non-uniform mesh. The time step size for the next time step n + 1 is dynamically updated according to the in the computational scheme. Since diffusive processes are dominant, we utilize the mesh Fourier number based on the diffusivity  $\frac{\beta_T}{\alpha_T}$  of heat of the current time step n

Since this choice for the time step computation did not yield instabilities, we excluded the vapor's diffusivity for the time step computation. The final iterative approach can be summarized as: determine time step size  $\Delta t$  according to Eq. , update the temperature field based on Eq. , compute the deposition rate with the temperature field based on Eq. , determine the vertical velocity with Eqs. Note that in response to settling processes, the mesh sizes vary and decrease (cf. Fig. 1), and , and update

390 the ice volume fraction and the mesh coordinates simultaneously based on Eqs. and . While 1.-3. is a re-implementation of an existing approach previously published by (Hansen and Foslien, 2015; Calonne et al., 2014), their coupling to 4. and 5. constitutes the novelties of our work, see also Fig. 2. Note that 4. is computed as part of 5. in the code. Our implementation is

**Figure 2.** Illustrates the computational workflow of one iteration. The state variables at time  $t^n$ , depicted on the left hand side are updated through the modules annotated as dashed boxes , and that are ordered diagonally in the centre of the figure. After each update the state variables at time  $t^{n+1}$  are retrieved. The equations of the modules are implemented into the computational model through the respective solution technique stated in the solid boxes on the top row. The computational steps are carried out from top to bottom. The equations for heat and vapor transport have already been implemented by Calonne et al. (2014) and Hansen and Foslien (2015) iterative approach can be summarized as: 1) determine time step size  $\Delta t$  according to Eq.A feedback- (27), 2) update the temperature field based on Eq. (23), 3) compute the ice volume fraction in deposition rate with the absence of a vertical velocity has been investigated in the companion paper (?) temperature field based on Eq. The modules for (24), 4) determine the vertical velocity with Eqs. (17) and for the coupled-(18), and 5) update of the ice volume fraction and the mesh coordinates simultaneously based on Eqs. (19) and (20). While 1)-3) is a re-implementation of an existing approach previously published by Hansen and Foslien (2015); Calonne et al. (2014), through their coupling to 4) and 5) constitutes the method novelties of characteristics our work. Note that 4) is novel computed as part of 5) in our approachthe code.

[revised manuscript text omitted]